



# Evaporation from northern latitude wetlands

Astrid Vatne[1], Norbert Pirk[1], Kolbjørn Engeland[1,2], Ane V. Vollsnes[3], and Lena M. Tallaksen[1]

[1]Department of Geosciences, University of Oslo
[2]Norwegian Water Resources and Energy Directorate
[3]Department of Biosciences, University of Oslo

**Correspondence:** Astrid Vatne (astrid.vatne@geo.uio.no)

**Abstract.** The atmospheric demand for evaporation in northern latitude ecosystems is expected to increase with increasing temperatures and a longer snow-free season. To understand how increased evaporative demand will affect ecosystems in this typically moisture-rich region, we need more knowledge about the factors that control evaporation and, furthermore, how evaporation modifies local hydrology. We used year-round evaporation estimates from four eddy-covariance wetland sites in

Norway to quantify evaporation and identify its main controls along climatic gradients in temperature and precipitation. We found that ecosystem evaporation was indeed mainly controlled by atmospheric evaporative demand and spring snow-cover duration. Soil moisture remained high during the measurement period and likely never reached a level where it would impact evaporation. Annual evaporation ranged from 81 mm to 208 mm and increased with warm-season mean temperature along the spatial gradient. We found a large variation in the role of evaporation in the ecosystem water balance, with annual evaporation

ranging from 9 % to 30 % of annual precipitation. In the warm season, evaporation was typically around 50 % of the seasonal precipitation, but reached a maximum of 72%. Compared to other northern latitude sites in the FLUXNET2015-dataset, the evaporation from the Norwegian sites was lower than what would be expected from the site warm season mean temperatures. Our results show that evaporation is an important part of the northern latitude water balance, especially during the warm season and in parts of the region with low precipitation. Furthermore, our results indicate that earlier snow-cover melt-out and

increased vapour pressure deficit have the potential to increase annual evaporation.

## 1 Introduction

There is a high demand for observations of evaporation in fast-warming northern latitude regions, which are characterised by a seasonal snow cover. These are typically moisture rich regions with numerous lakes and wetlands. Evaporation is in general energy-limited, i.e., the evaporation is mainly controlled by the amount of available energy (McVicar et al., 2012), rather

than water-limited, and it is traditionally considered a minor component of the annual water balance. As the climate warms, snow cover generally reduces and the length of the growing season increases (Rizzi et al., 2017). Further, it is expected that the atmospheric demand for water will increase as air temperature increases (Masson-Delmotte et al., 2018; Grossiord et al., 2020).

The term "atmospheric evaporative demand" is used to quantify the combined effect of the available energy for evaporation

and the ability of the atmosphere to receive water vapour, and is typically used interchangeably with "potential evaporation"





(Peng et al., 2018). The atmospheric demand typically increases with increasing net radiation, vapour pressure deficit and wind speed (Katul et al., 2012). Vapour pressure deficit affects the partitioning of surface energy into sensible and latent heat fluxes. Liljedahl et al. (2011) found a vapour pressure deficit of 0.3 kPa to be an important threshold, favouring latent over sensible heat flux for two drained thaw lake basins on the Arctic Coastal Plain of Alaska. The snow cover can influence the evaporative demand through its effect on the net radiation, due to the high albedo of snow compared to snow-free ground. Changes in the length of the snow-free season and an increased vapour pressure deficit can cause changes in the water balance of northern latitude ecosystems by affecting the terrestrial evaporation. Pirk et al. (2023b) found that annual total evaporation from an alpine tundra site decreased by 50 % in a year with a one-month delayed melt-out date of the snow-cover.

To better understand how northern latitude ecosystems respond to a longer snow-free season and an increased evaporative demand, we need more knowledge on the magnitude and controls of evaporation. Observations of evaporation are here key, both to increase process understanding and to constrain models. Existing hydrological models show large spread in evaporation estimates, ranging from 178 mm to 500 mm annually for mainland Norway (Erlandsen et al., 2021). Routine observations of evaporation in Norway are lacking, and although evaporation data from eddy-covariance measurements are available through networks such as FLUXNET (Pastorello et al., 2020), existing sites and previous studies have often focused on forested areas. However, land cover types such as wetland and tundra, are widespread in this region. Current knowledge on the magnitude and dynamics of evaporation in these ecosystems is highly uncertain. Time series of observations are available from wetland and forest sites in e.g. Finland and Sweden, but due to the large climatic gradients in temperature and precipitation, it is uncertain if existing observations are representative of ecosystems in the larger region. Though evaporation has been measured at a limited number of eddy covariance sites across the Northern Hemisphere, its analysis has received less attention than carbon fluxes (Baldocchi, 2020).

This study explores how evaporation and its main environmental controls vary with climatic gradients of temperature and precipitation at northern high latitudes. We use new observations from three wetland and tundra sites in mainland Norway, a previously poorly represented region (Pallandt et al., 2022). We also include a site on Svalbard to cover a larger climatic gradient. We aim to quantify evaporation at sub-daily to annual time scales. Furthermore, we explore to what extent available energy controls evaporation, and what the role of other factors, such as vapour pressure deficit, soil moisture and snow cover, plays in controlling evaporation rates. Our main objective is to quantify evaporation from northern latitude wetlands in Norway and identify its main climatic controls. More specifically, we look at (1) the controls on evaporation at an hourly timescale, (2) the magnitude and seasonality of daily and monthly evaporation rates and (3) the annual evaporation and interannual variability across climatic gradients. To place the new sites in a regional context, we compare the annual evaporation of our sites to that of existing northern latitude sites in the FLUXNET2015-dataset.



**Table 1.** Site information, mean annual precipitation (MAP), and mean annual air temperature (MAAT) for the climate reference period 1991-2020 (based on data from The Norwegian Meteorological Institute).

| Site | Latitude | Longitude | Altitude (m.a.s.l.) | MAP (mm) | MAAT (°C) | Measurement period |
|------|----------|-----------|---------------------|----------|-----------|--------------------|
| Hisåsen | 61.11N | 12.25E | 640 | 857 | 2.7 | 1.1.2020 - 31.12.2022 |
| Finse | 60.59N | 7.53E | 1210 | 968 | -1.1 | 1.1.2019 - 31.12.2022 |
| Iškoras | 69.34N | 25.30E | 380 | 417 | -1.4 | 24.3.2019 - 31.12.2021 |
| Adventdalen | 78.19N | 15.92E | 14 | 218 | -3.9 | 1.1.2013-31.12.2013 and 1.1.2015-31.12.2016 |

## 2 Materials and Methods

### 2.1 Study sites

The four study sites are Finse, Hisåsen and Iškoras on mainland Norway and Adventdalen on Svalbard. The sites are located along gradients in latitude, temperature and precipitation, covering a latitudinal gradient from 60–78 °N, a precipitation gradi-
ent from 218–968 mm per year and a gradient in mean temperature from -3.9–2.7 °C (see Figure 1 and Table 1). The vegetation at the three mainland sites has been classified according to the 'Nature in Norway' ecosystem and landscape diversity framework (Halvorsen et al., 2020). Compared to northern latitude (above 60 °N) sites available in the FLUXNET2015 dataset (Pastorello et al., 2020), our study sites span approximately the full range of latitudes and annual precipitation rates (including one site with higher precipitation), whereas they are somewhat in the mid-range with respect to mean annual temperature
(Figure 1b).

The *Hisåsen* site is a boreal peatland site, located south of the hill Hisåsen in the Regnåsen-Hisåsen nature reserve in eastern Norway. The area is undulating, slightly sloping towards north and covered by forest and peatlands. The climate is boreal, according to Köppen's classification, with a mean annual temperature of 2.7 °C and a mean annual precipitation of 857 mm. The measurement tower is located on a drained peatland with organic soils surrounded by forest on glacial till (NGU). The
site is part of a wetland restoration project and is instrumented with two twin measurement towers, where one tower measures gas exchange on a mire that was restored in 2021 and the other tower measures gas exchange over a control drained mire. Only the control mire is included in this study, as the ecosystem has remained untreated during the measurement period. The ecosystem types in the area have been mapped in the Nature in Norway system (Halvorsen et al., 2020) and maps are publicly available (github.com/geco-nhm/NiN_Hisaasen), showing that the footprint is dominated by strongly lime-poor fen
and lime-poor drained fen.

*Finse* is a sub-alpine tundra and wetland site, located in the valley Finsedalen north of the Hardangerjøkulen glacier. The valley runs towards east-southeast, and wind directions at the site are controlled by the valley. The climate is alpine, with mean annual temperature of -1.1 °C and mean annual precipitation of 967 mm. The climate has an oceanic influence, as the site is located approximately 140 km from the coastline in the west. The instrument tower sits on a ridge running southwest-northeast.
Towards southeast, the ridge slopes down towards the river Ustekveikja draining the lake Finsevatnet, located approximately





a)

b)



**Biome**

N/A
Temperate Broadleaf & Mixed Forests
Boreal Forests/Taiga
Tundra
Temperate Conifer Forests

**FLUXNET 2015**

● Forest
● Wetland
○ Other ecosystems

**Study sites**

● Finse
● Hisåsen
● Iškoras
● Adventdalen

**Figure 1.** Location (a) and climatic context (b) of the four sites included in the study compared to selected northern latitude sites from FLUXNET2015 (Pastorello et al., 2020). Finse in blue, Hisåsen in green, Iškoras in orange, Adventdalen in red and FLUXNET sites in greyscale. The symbols of the FLUXNET sites indicates if the site is a wetland (black), evergreen need leaf forest (dark grey) or other ecosystem types (light grey). The latter category includes one site in each of the following ecosystem types: open shrubland, cropland, grassland, and snow/ice. Only FLUXNET sites located above 60 °N latitude and with Creative Commons (CC-BY-4.0) licence were included in the comparison (see a list of the included sites in Supporting table B1). The biome map in panel a) is from Dinerstein et al. (2017).

1 km east of the tower. The footprint vegetation types are predominantly lime-poor open fens, arctic-alpine heath and lee side as well as snowbeds according to the Nature in Norway system (Bryn, 2020; Halvorsen et al., 2020). The soil is thin and consisting of discontinuous glacial till with thickness less than 0.5 m and glacifluvial deposits (NGU).

*Iškoras* is a palsa mire site located north of the mountain Iškoras in the plateau area of Finnmarksvidda in northern Norway. 85 The climate is subarctic, with a mean annual temperature of -1.4 °C and a mean annual precipitation of 417 mm. The tower is



located on a peat plateau surrounded by mires and ponds (Martin et al., 2019), with partly organic soil and partly glacial till (NGU). Shrubs and lichens dominate dry, elevated palsas, whereas sedges and mosses dominate wetter areas near unvegetated ponds (Pirk et al., 2023a). A vegetation mapping found bog and open fen to be the dominating types (Halvorsen et al., 2020; Bryn, 2023). The peat plateau lies north of the mountain range Iškoras, and the terrain slopes gently towards north.

*Adventdalen* (NO-Adv) is an Arctic site featuring polygonal tundra, located in the valley Adventdalen on Spitsbergen, Svalbard. The climate is polar, with a mean annual temperature of -3.9 °C and a mean annual precipitation of 218 mm. The climate has an oceanic influence, as the site is located approximately 6 km from the coast. The tower is located on a river terrace on the flat part of a large alluvial fan, and the soil consists of a few decimetres of fine-grained eolian deposits on top of coarser-grained alluvial deposits (Pirk, 2017). The vegetation is very low, dominated by dwarf shrubs like *Salix polaris* at dry
places and mosses and sedges in wet depressions.

     For each site, the 3–4 years with the most complete data collection is chosen as the site measurement period. Thus, the measurement period varies slightly between sites (Table 1). Figure 2 shows monthly mean temperature and cumulative precipitation at the nearest stations of The Norwegian Meteorological Institute (MET Norway) for the reference period (1990-2020) and for each year in the respective measurement periods. Precipitation is relatively evenly distributed over the year at all sites,
with the wettest month having a mean precipitation of 2.7 to 3.6 times the precipitation of the driest month. The wettest month of each site occurs in summer and early autumn (July at Iškoras, August at Hisåsen and September at Finse and Adventdalen). The driest month occurs in spring to early summer (March at Hisåsen and Iškoras, April at Finse and May at Adventdalen). Monthly mean temperature typically peaks in July and is below zero from November-March at Hisåsen, October-April at Finse and Iškoras and October-May at Adventdalen.

Overall, the measurement years were warmer than the reference period at all sites (Figure 2). At Hisåsen, the annual mean temperature was warmer than normal for all years (0.1 to 0.2 °C), with 2020 being the warmest year. Annual precipitation was both higher and lower than normal (ranging from 115 % to 88 % of the long term mean) with 2020 being the wettest year. A similar pattern was found at Finse, with all years being warmer than normal (0.1 to 1.0 °C), and annual precipitation both higher and lower than normal (ranging from 78 % to 133 % of the long term mean), with 2020 being the wettest year.
Iškoras was the only site where annual temperature was both higher and lower than normal (varying from 0.4 °C lower in 2019 and 2021, to 0.7 °C higher in 2020). However, all years had temperatures 0.2 °C to 4.6 °C above normal in the warm season (May–September). Annual precipitation ranged from 101 % to 118 % of the long term mean, with 2019 and 2020 being the wettest years. At Adventdalen, all years were warmer (0.4 to 3.8 °C) and wetter (114 % to 143 %), than the long term mean, with 2016 being the warmest and wettest year as well as the warmest ever recorded at the station. Annual precipitation ranged
from 114 % of the long term mean in 2013 to 143 % in 2016. The temperature deviations were typically larger in autumn and winter (September–March) than in spring and summer (April–August).





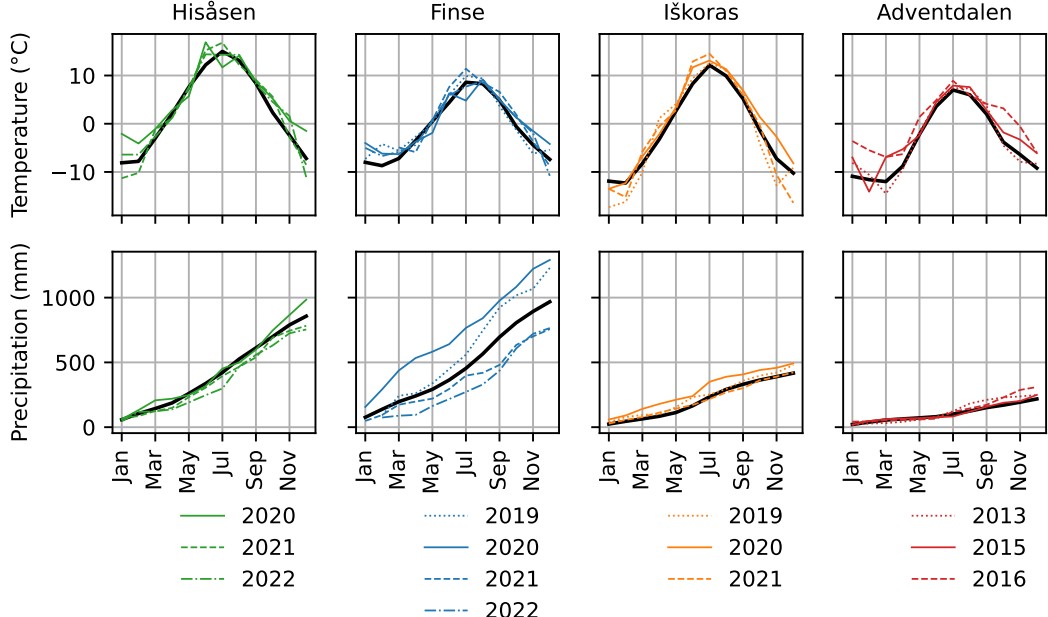

**Figure 2.** Monthly mean temperature (upper row) and cumulative precipitation (lower row) in climate reference period (black lines) and measurement periods (coloured lines) at each site's nearest MET Norway weather station.

## 2.2 Data sources and processing

### 2.2.1 Eddy covariance data

To measure ecosystem evaporation, we used the eddy covariance method (e.g., Burba, 2022). At all sites, we measured water vapour mixing ratio using an enclosed gas analyser (Li-Cor LI7200). Three-dimensional wind speed measurements were made by sonic anemometer (Campbell Sci. CSAT3 at Finse and Iškoras, Gill HS-50 at Hisåsen, and a Metek USA-1 at Adventdalen). We processed the raw eddy covariance data in the software EddyPro version 6.2.0, using a double rotation tilt correction of the anemometer, block average method to extract turbulent fluctuations and constant time lag between wind and gas concentration based on pump flow rate. We applied spectral corrections in the high frequency range according to Moncrieff et al. (1997), and in the low frequency range according to Moncrieff et al. (2005). The resulting time series of half-hourly evaporation flux was filtered according to the 0–2 flagging scheme based on tests proposed in Foken and Wichura (1996), discarding all observations with quality flags >0. Additionally, the time series of fluxes and ancillary variables measured by the eddy covariance system (friction velocity and wind speed) were filtered based on a statistical screening of the raw data, using tests from Vickers and Mahrt (1997) with the default test thresholds in Eddypro 6.2.0 (hard-flags only).



To estimate daily, monthly and annual evaporation, we filled the gaps in evaporation time series by building a random forest regression model (Python package Sklearn) for each site, using gap-filled ancillary data as predictors (Table 2 and 3). The observed evaporation was averaged to hourly mean values. Before building the regression, the dataset was split into 75 % training and 25 % test data. The root-mean-square error (RSME) of test data predictions varied from 0.011 mm h$^{-1}$ to 0.015 mm h$^{-1}$.

Due to the large variation in surface cover in the footprint area of the Finse site, we grouped the data based on the dominating wind direction (east and west) and gap-filled the two wind direction sectors separately (as done in Pirk et al. (2023b)). Only data from the western sector were included in this study, as the western surface cover is more comparable to the other sites in the study.

### 2.2.2    Measured ancillary local data

We used locally measured meteorological and surface variables to gap-fill the evaporation time series and to identify controls on evaporation on a sub-daily timescale (hourly values). At each site, measurements of local meteorological and surface variables (Table 2) were sampled with one minute time resolution. We discarded periods of data with sensor error through visual inspection. After filtering, the values were aggregated to hourly means. Gaps in the time series of meteorological variables (air temperature, vapour pressure deficit, wind speed, incoming radiation and atmospheric pressure) were filled using the

bias-corrected corresponding ERA5 Land variable (Muñoz Sabater, 2019), downloaded from the Climate Data Store (CDS, accessed on 22 Feb 2023) of the Copernicus Climate Change Service (C3S). We bias-corrected the ERA5 Land variables by using a simple linear regression with the corresponding local variable, built by using either data from the whole year or, when data coverage for each season was sufficient, by building seasonal linear regressions for winter (December-February), spring (March-May), summer (June-August) and autumn (September-November) separately. Vapour pressure deficit was derived from

relative humidity and temperature. We created a gap-free time series of hourly precipitation using data from the nearest MET station (Supporting table A1), gap-filled with ERA5 Land precipitation. The hourly time series of precipitation was used to derive a new variable called "time since rain" (as described in section 2.2.3), that we used as a proxy of soil moisture availability, in addition to point measurements of soil water content where available. The time since rain variable was considered to be more representative of a larger area than point measurements. To calculate monthly and annual evaporation ratio (evaporation as fraction of precipitation) we used monthly and annual precipitation from nearest MET Norway station (Supplementary table

A1).

We gap-filled ancillary surface variables (outgoing radiation, soil temperature, soil water content and soil heat flux) by using predictions from a random forest regression (Python package Sklearn). A random forest regression was built for each variable, using gap-filled meteorological variables and derived variables (table 3) time since rain, growing degree days and snow cover

as predictors.



**Table 2.** Measured ancillary data

| Variable | Abbreviation | Source | Gap filled by |
|---|---|---|---|
| Air temperature | TA | Measured locally | Bias corrected ERA5 Land air temperature |
| Air relative humidity | RH | Measured locally | Bias corrected ERA5 Land air relative humidity |
| Air pressure | PA | Measured locally | Bias corrected ERA5 Land air pressure |
| Wind speed | WS | Measured locally | Bias corrected ERA5 Land wind speed |
| Precipitation | P | Nearest MET station [a] | ERA5 Land precipitation |
| Shortwave incoming radiation | $SW_{in}$ | Measured locally | Bias corrected ERA5 Land incoming shortwave radiation |
| Longwave incoming radiation | $LW_{in}$ | Measured locally | Bias corrected ERA5 Land incoming longwave radiation |
| Shortwave outgoing radiation | $SW_{out}$ | Measured locally | Estimates from random forest regression model |
| Longwave outgoing radiation | $LW_{out}$ | Measured locally | Estimates from random forest regression model |
| Soil temperature | TS | Measured locally | Estimates from random forest regression model |
| Soil heat flux | SHF | Measured locally | Estimates from random forest regression model |
| Soil volumetric water content [b] | SWC | Measured locally | Estimates from random forest regression model |
| Water table depth [c] | WTD | Measured locally | Estimates from random forest regression model |

[a] At sites Hisåsen and Iškoras, precipitation is measured locally in the snow-free season and from the nearest MET Norway station in the snow-covered season.

[b] ) Not measured at Adventdalen.

[c] Only measured at Hisåsen.

### 2.2.3  Derived ancillary data

To provide additional information for the gap-filling of surface variables and evaporation fluxes, we derived variables representing snow cover, soil moisture availability (time since rain) and phenology (growing degree days). Time since rain was calculated from the gap-filled time series of hourly precipitation. If hourly precipitation exceeded 0.1 mm, we defined it as a
rain event. For each time step, we then calculated hours passed since the last rain event. In the snow-covered season, we set time since rain to zero. To calculate growing degree day, we used the gap-filled time series of hourly temperature. Growing degree day was then calculated according to its usual definition.

The ground surrounding the towers, approximately $1 \, \mathrm{km}^2$, was classified as either i) snow-free, ii) partly snow-covered or iii) fully snow-covered, by visually inspecting satellite images (Sentinel-2, natural colour, accessed through Copernicus Browser
(Copernicus Data Space Ecosystem)) during spring and autumn each year for each station. The start/end of the snow-free season was set to the date of the first/last available image of snow-free ground. Similarly, the start/end of the snow-covered season was set to the date of the first/last image with a full snow cover that lasted through the winter. The period in between was considered as a shoulder season, with either partly snow-covered ground or a snow cover lasting only for a short while. Controls in the snow-free season were analysed by masking data from days when the ground was either fully or partly snow-covered.






**Table 3.** Derived ancillary data

| Variable | Abbreviation | Source |
|---|---|---|
| Vapour pressure deficit of air | VPD | Derived from TA and RH |
| Surface albedo | albedo | Derived from $SW_{in}$ and $SW_{out}$ |
| Time since rain | TSR | Derived from P |
| Growing degree day | GDD | Derived from TA |
| Snowfree season | snowfree | Determined visually from Sentinel 2 images |
| Shoulder season | shoulder | Determined visually from Sentinel 2 images |
| Available energy[a] | $R_n - G$ | Derived from $SW_{in}$, $SW_{out}$, $LW_{in}$, $LW_{out}$, and SHF. |

[a] Used as forcing for Penman-Monteith equation (described in section 2.3.2). Not included in gap-filling or factor analysis.

## 2.3 Identifying controls on evaporation

To study the controls on evaporation on a sub-daily timescale, we used hourly observations of evaporation from the eddy covariance data. Only observed data were used in this part of the study (i.e. gap-filled values were not included). We first calculated Pearson correlation coefficients between hourly evaporation and local meteorological and surface variables for the snow-free and snow-covered season separately, and tested whether the correlation coefficients were significant at significance levels $p<0.05$, $p<0.01$ and $p<0.001$. Available energy ($R_n$-G) is included in the analysis of correlations as it is a forcing variable in the Penman-Monteith equation (described in section 2.3.2), but is not included as a predictor variable in the gap-filling regression model or included in the factor analysis. To avoid spurious predictor importances due to the large degree of covariance between the controls of evaporation, we performed a Factor analysis (similar to the analysis in Thunberg et al. (2021b) and Thunberg et al. (2021a)) to group variables with a large degree of common variance. To evaluate how the partitioning between sensible and latent heat flux changed with atmospheric and surface controls, we calculated mean Bowen ratio, i.e. the ratio of sensible to latent heat flux, for bins of vapour pressure deficit and soil water content. Finally, we modelled hourly evaporation using the Penman-Monteith equation to test how a widely used model is able to capture the sensitivity of evaporation to climatic and surface controls. We focus on controls in the snow-free season, as 68 % to 86 % of the annual evaporation occurred in the snow-free season.

### 2.3.1 Factor analysis

To group variables with common variability pattern, and identify control variables with a high degree of common variance as evaporation, we performed an exploratory factor analysis using the Python package 'Factor Analyzer'. The factor analysis groups the observed variables into underlying unobserved variables called factors. Each factor explains a certain variance in the dataset of observed variables, with the first factor explaining the most. The result of the analysis can be interpreted by the observed variable's factor loading, i.e. the correlation coefficient between a factor and an observed variable. Observed variables with a high degree of common variance will load high on the same factor. For each site and each season (snow-free and snow-





covered), we first evaluated the suitability of the dataset for factor analysis using the Kaiser-Meyer-Olkin (KMO) criterion
(Cureton and D'Agostino, 2013), and performed the analysis only if the dataset KMO-value exceeded 0.5. The number of
factors was based on the Kaiser criterion, with an eigenvalue of 1 as threshold. For the final factor extraction, we used 'Varimax'
orthogonal rotation, which seeks to minimize the number of variables that have high loading on each factor. We specifically
looked at variables with a high degree of loading on the same factor as evaporation.

### 2.3.2   Penman-Monteith Estimates

Hourly evaporation in the snow-free season was modelled from the Penman-Monteith equation (Monteith, 1965) as:

$$E_{pm} = \left(\frac{3.6 \cdot 10^6}{\rho_w \lambda_v}\right) \frac{\delta(R_n - G) + \rho_a c_p g_a VPD}{\delta + \gamma\left(1 + \frac{g_a}{g_s}\right)} \tag{1}$$

where $E$ is evaporation rate in mm h$^{-1}$, $\rho_w$ is the mass density of water in kg m$^{-3}$, $\lambda_v$ is the latent heat of vaporisation in J
kg$^{-1}$, $R_n - G$ is available energy in W m$^{-2}$, $\gamma$ is the psychrometric constant in Pa °C$^{-1}$, $\delta$ is the slope of the saturation vapour
pressure versus temperature curve in Pa °C$^{-1}$, $\rho_a$ is the mass density of dry air in kg m$^{-3}$, $c_a$ is the specific heat of air in J
°C$^{-1}$, $VPD$ is air vapour pressure deficit in Pa, $g_s$ is surface conductance and $g_a$ is aerodynamic conductance, the latter two
in m s$^{-1}$.

As forcing for the Penman-Monteith equation, we used the gap-filled ancillary data (Table 2). Available energy was estimated
as the difference between net radiation and the soil heat flux, i.e. as:

$$R_n - G = SW_{in} - SW_{out} + LW_{in} - LW_{out} - SHF \tag{2}$$

Aerodynamic conductance was estimated from average wind speed, $WS$, in m s$^{-1}$:

$$g_a = \frac{WS}{\left(6.25 \log \frac{z}{z_0}\right)^2} \tag{3}$$

Where $z$ is wind speed measurement height and $z_0$ is the surface roughness length (estimated by visual inspection of vegetation
height, see Table 4), both in m. We assumed the zero plane displacement to be zero, as the vegetation height is low at all sites.
We used a site-specific constant surface conductance parameter $g_s$. An optimal $g_s$ value for each site was derived by choosing
the value that minimized the root square mean error (RMSE) in an interval of $g_s$ from 0 m s$^{-1}$ to 0.05 m s$^{-1}$, covering the
parameter range found for high latitude ecosystems in Kasurinen et al. (2014), with increments of 0.0001 m s$^{-1}$. Parameter
values of $z$, $z_0$ and $g_s$ at each site are listed in Table 4.

To evaluate the sensitivity of the observed and estimated evaporation to various controls, we studied the relative error of
the Penman-Monteith $E_{pm}$ estimate to the observed evaporation $E_{obs}$. The relative error was calculated as $\frac{E_{pm} - E_{obs}}{|E_{obs}|}$. We
evaluated the sensitivity to the main forcing variables by looking for patterns in the mean relative error for bins of available
energy and vapour pressure deficit. Furthermore, we evaluated the appropriateness of using a constant surface conductance
parameter. The surface conductance parameter represents how available water is for evaporation on the surface, and can be seen
as a combination of stomata conductance, leaf area index and soil water conductance. The surface conductance is typically





**Table 4.** Parameters used in the Penman-Monteith equation for estimating hourly evaporation.

|  | Hisåsen | Finse | Iškoras | Adventdalen |
|---|---|---|---|---|
| $z$ (m) | 2.8 | 4.4 | 2.8 | 2.8 |
| $z_0$ (m) | 0.30 | 0.06 | 0.04 | 0.01 |
| $g_s$ (m s$^{-1}$) | 0.0026 | 0.0009 | 0.0010 | 0.0010 |

modelled as a function of phenology, soil water content, solar radiation and temperature (e.g. Stewart, 1988). To evaluate the effect of soil moisture content and phenology on evaporation, we used a constant surface conductance and analysed the

deviation from observed evaporation by looking at the tendency of the mean relative error to change for bins of the variables volumetric soil water content, time since rain and growing degree day. For volumetric soil water content, we used 20 bins in the range of observed values at each site, since the numeric value is not directly comparable between sites due to differences in soil properties. For time since rain and growing degree days, we used 24 h bins and 90 °C bins, respectively. For each bin, any data point where the relative error was more than 1.5 times the interquantile range below the first quantile or above the third

quantile, was considered an outlier and removed. Only bins with a minimum of 10 data points remaining were used in analysis.

### 2.4 Regional comparison

To evaluate how the evaporation at the sites in this study compares to that of other northern latitude sites, we compared mean annual evaporation of the four sites to those in the FLUXNET2015 dataset (Pastorello et al., 2020). Only FLUXNET sites located above 60 °N latitude with Creative Commons (CC-BY-4.0) licence were included in the comparison (see a list of

the sites in Supporting Table B1). To test the annual evaporation dependency on the site mean temperature, we fitted a linear regression with annual evaporation as dependent variable and mean annual temperature as independent variable and checked if the regression slope was significant at p<0.05. Furthermore, we tested if a better fit could be obtained by using warm season (May-September) mean temperature only instead of annual mean temperature.

### 3 Results

### 3.1 Controls on evaporation

Hourly evaporation was significantly correlated (p<0.001) with most of the climatic and surface controls, both in the snow-free and snow-covered season (Table 5), however, the relation was typically weak (r<0.5) except for a few controls related to atmospheric evaporative demand. In the snow-free season, evaporation had strong linear relations (r>0.7) to vapour pressure deficit and incoming shortwave radiation. Most stations had a strong or moderate relation (r>0.5) to air and soil temperature and

outgoing longwave radiation, while the relation to other variables such as soil water content and wind speed was weak. In the snow-covered season, the correlation coefficients between evaporation and its controls were overall weaker than in the snow-free season, however the relation with incoming shortwave radiation and vapour pressure deficit was still strong or moderate





**Table 5.** Pearson correlation coefficients between hourly evaporation (ET) and climatic and surface controls in both snow-free season and snow-covered season (see Table 2 and 3 for abbreviations).

| | Hisåsen | | Finse | | Iškoras | | Adventdalen | |
|---|---|---|---|---|---|---|---|---|
| Variable | Snow-Free | Snow-Covered | Snow-Free | Snow-Covered | Snow-Free | Snow-Covered | Snow-Free | Snow-Covered |
| TA | 0.67*** | 0.45*** | 0.67*** | 0.18*** | 0.64*** | 0.33*** | 0.19*** | 0.32*** |
| VPD | 0.81*** | 0.72*** | 0.81*** | 0.63*** | 0.80*** | 0.77*** | 0.70*** | 0.58*** |
| WS | -0.22*** | -0.03 | -0.27*** | -0.18*** | 0.01 | -0.04 | 0.08*** | 0.13*** |
| PA | 0.26*** | 0.12*** | 0.20*** | 0.16*** | 0.15*** | 0.21*** | 0.20*** | 0.05* |
| $R_n$-G | 0.89*** | 0.62*** | 0.72*** | 0.45*** | 0.82*** | 0.65*** | 0.76*** | 0.16*** |
| SWIN | 0.92*** | 0.69*** | 0.80*** | 0.51*** | 0.86*** | 0.70*** | 0.79*** | 0.32*** |
| albedo | -0.06*** | -0.23*** | 0.06** | -0.16*** | -0.06*** | -0.29*** | -0.01 | -0.29*** |
| LWIN | 0.15*** | 0.06*** | -0.19*** | -0.20*** | 0.05** | -0.01 | -0.42*** | 0.21*** |
| LWOUT | 0.82*** | 0.38*** | 0.84*** | 0.21*** | 0.82*** | 0.44*** | 0.47*** | 0.32*** |
| SHF | 0.73*** | 0.02 | 0.52*** | 0.20*** | 0.71*** | 0.31*** | | |
| TS | 0.48*** | 0.00 | 0.50*** | 0.08*** | 0.63*** | 0.31*** | 0.40*** | 0.32*** |
| SWC | -0.31*** | 0.22*** | -0.23*** | -0.1*** | -0.19*** | -0.30*** | | |
| TSR | 0.23*** | 0.08*** | 0.31*** | 0.18*** | 0.21*** | 0.05 | 0.14*** | -0.04 |
| GDD | -0.45*** | -0.26*** | -0.29*** | -0.18*** | -0.32*** | -0.35*** | -0.31*** | -0.09*** |

\* Significant at $p < 0.05$.

\*\* Significant at $p < 0.01$.

\*\*\* Significant at $p < 0.001$.

(r>0.5) for all sites except at Adventdalen where only vapour pressure deficit had r > 0.5 in the snow-covered season. For other controls, the relation to evaporation was weak (r<0.5).

The factor analysis showed that evaporation, vapour pressure deficit and incoming shortwave radiation had a high degree of shared variance both in the snow-free and snow-covered season. In the snow-free season, evaporation loaded highest on the first factor at Hisåsen, Finse and Iškoras (Supporting Figure C3), together with vapour pressure deficit, incoming shortwave and outgoing longwave radiation (all with loadings >0.72). Air temperature, soil temperature and soil heat flux also loaded relatively high (>0.55) on the first factor. At Adventdalen, evaporation loaded highest on the second factor, together with incoming
shortwave radiation and vapour pressure deficit (>0.62), whereas variables such as air temperature, surface temperature, and longwave outgoing radiation loaded high on the first factor. In the snow-covered season, evaporation had a lower loading on the first and second factor compared to the snow-free season, indicating a lower degree of common variance with other variables in the dataset (Supporting Figure C4). At Finse and Adventdalen, the loading for evaporation was less than 0.51. At Hisåsen and Iškoras however, the evaporation loadings in the snow-covered season was more comparable to the snow-free season (>0.7).
At Hisåsen, evaporation had a relative high loading (0.76) on the first factor, together with shortwave incoming radiation (0.79)



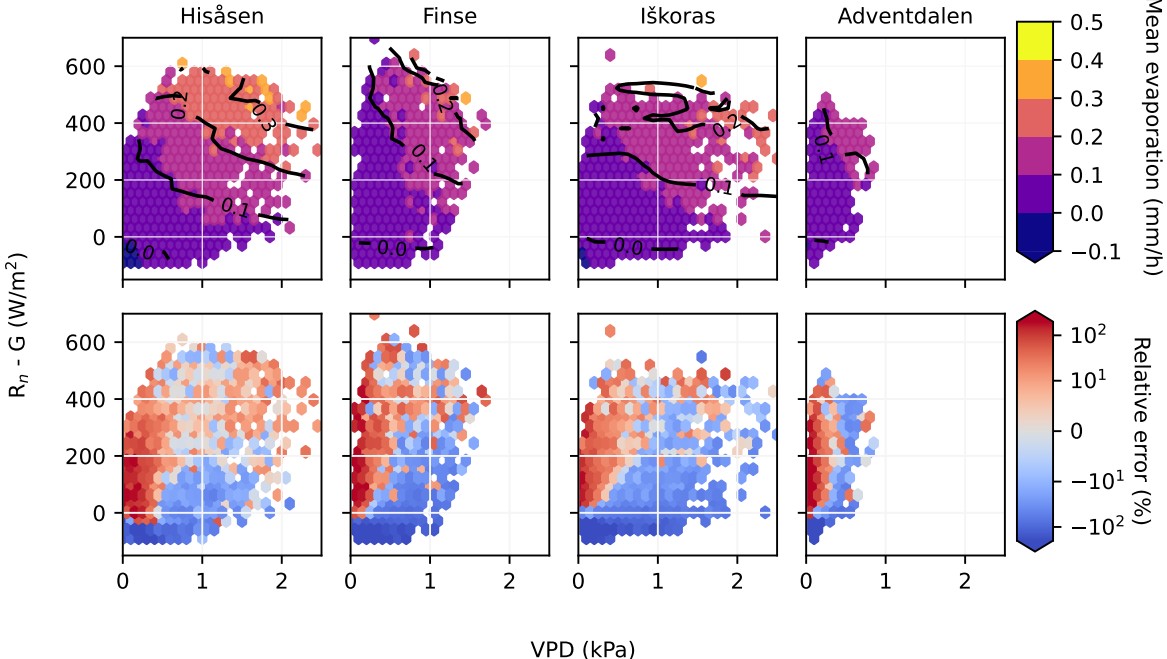

**Figure 3.** Upper row shows mean evaporation (mm h$^{-1}$) for bins of available energy (W m$^{-2}$) and vapour pressure deficit (kPa) for observed evaporation in colour plot, and modelled evaporation in contour plot (solid black lines). The lower row shows the relative error (%) of the modelled evaporation by the Penman-Monteith equation for the same bins of available energy and vapour pressure deficit.

and vapour pressure deficit (0.90). At Iškoras, evaporation had a relative high loading on the second factor (0.84) together with incoming shortwave radiation (0.76) and vapour pressure deficit (0.71).

    Comparing observed evaporation to Penman-Monteith estimates in the snow-free season, and looking at the distribution for bins of available energy and vapour pressure deficit (Figure 3, upper row), we found that the Penman-Monteith equation

reproduced the observed pattern of high evaporation when both vapour pressure deficit and available energy were high, and low evaporation when both controlling factors were low. The mean relative error (Figure 3, lower row) showed a similar pattern across the sites, and showed a tendency towards overestimation (red hexagons) when vapour pressure deficit was low, and underestimation (blue hexagons) when high vapour pressure deficit was combined with low available energy. Furthermore, evaporation was underestimated by the Penman-Monteith equation when available energy was negative.

We evaluated the sensitivity of the observed and estimated evaporation, and the relative error of the estimates, to surface controls that can affect the surface water availability, such as the soil water content and time since rain (Supporting figure C5-C6), and growing degree day which is related to phenology (Supporting figure C7). Overall, the sensitivity of the Penman-Monteith estimates corresponded to the observed sensitivity, and there was no tendency for the relative error to increase or decrease with changes in the surface controls. However, there was a tendency towards higher relative errors at high values of

soil water content at Finse and Hisaasen (Supporting figure C5).



The Bowen ratio, i.e. the ratio of sensible to latent heat flux, decreased with vapour pressure deficit (Supporting figure C8), meaning that latent heat flux was increasingly favoured over sensible heat flux with increasing vapour pressure deficit. At vapour pressure deficit over 0.4 kPa to 1.0 kPa, the Bowen ratio was under 1 and the latent heat flux dominated over sensible heat. At Hisåsen, the mean Bowen ratio stabilised at just below 1 for vapour pressure deficit over 1.0 kPa, while at the other
sites, it continued to decrease for the whole range of observed values of vapour pressure deficit.

### 3.2 Magnitudes of evaporation

Daily evaporation rates showed a clear seasonal pattern, with lower values in the snow-covered season and higher values in the snow-free season. Across the sites, mean daily evaporation (averaged over season) ranged from 0.0 mm d$^{-1}$ to 0.1 mm d$^{-1}$ in the snow-covered season, from 0.2 mm d$^{-1}$ to 0.4 mm d$^{-1}$ in the shoulder season and from 0.5 mm d$^{-1}$ to 1.0 mm d$^{-1}$ in
the snow-free season. In the shoulder and snow-free season, the magnitudes of mean daily evaporation followed the gradient in mean annual temperature, with the highest evaporation at Hisåsen and lowest at Adventdalen, while in the snow-covered season there were only minor differences between the sites. Looking at the distribution of daily evaporation per month of the year (Figure 4, upper row), we found that the mean daily evaporation was highest in the summer months, June–August, at all sites. However, the month of the highest mean daily rates were slightly different between the sites. The mean daily evaporation
peaked in June at Hisåsen and Adventdalen, June/July at Iškoras and July/August at Finse. Comparing across sites, evaporation was highest at Hisåsen, with mean daily evaporation of 1.9 mm d$^{-1}$ (June), compared to 0.8 mm d$^{-1}$ at Finse (July/August), 1.2 mm d$^{-1}$ at Iškoras (June/July), and 0.8 mm d$^{-1}$ at Adventdalen (June).

The seasonal pattern of available energy and vapour pressure deficit followed a similar seasonal pattern, with lower values in winter and higher during summer (Figure 4, middle and lower row). Mean daily available energy ranged from -1.6 MJ d$^{-1}$
to 0.6 MJ d$^{-1}$ in the snow-covered season, from 1.2 MJ d$^{-1}$ to 5.0 MJ d$^{-1}$ in the shoulder season, and from 6.1 MJ d$^{-1}$ to 7.0 MJ d$^{-1}$ in the snow-free season. Mean daily vapour pressure deficit ranged from 0.04 kPa to 0.12 kPa in the snow-covered season, from 0.03 kPa to 0.15 kPa in the shoulder season, and from 0.14 kPa to 0.35 kPa in the snow-free season. Available energy peaked in June for all sites, while vapour pressure deficit peaked in either June, July, or August depending on site.

Across the sites, we found a large variation in the role of evaporation in the local water balance, due to a higher variation in
precipitation (Table 1) than in evaporation. The evaporation ratio, i.e. the ratio of evaporation to precipitation, was highest at Adventdalen, the site with the lowest precipitation, and lowest at Finse, the site with the highest precipitation. For individual months, the evaporation ratio was occasionally over 100 % at all sites, and up to 400 % at Adventdalen (Figure 5). At Finse, the monthly evaporation ratio was generally below 40 %, except in August 2021 when evaporation was 124 % of the precipitation. At Adventdalen, the monthly evaporation ratio was typically higher than 100 % in May, June, and July. Considering the warm
season (May-September), the total evaporation was up to 72 % of the precipitation in the same months (at Adventdalen in 2015). The mean warm season evaporation ratio was lowest at Finse (20 %), intermediate at Hisåsen (48 %) and Iškoras (47 %) and highest at Adventdalen (58 %). The mean annual evaporation ratio ranged from 9 % to 30 % of MAP (Figure 5), and the evaporation ratio increased with decreasing MAP across sites.



**Figure 4.** Daily evaporation in mm d$^{-1}$ (upper row), available energy in MJ m$^{-2}$d$^{-1}$ (middle row) and vapour pressure deficit in kPa (lower row) for each month at the four sites. For each month, the box plot represents median, 25- and 75 quantiles, and whiskers represent minimum and maximum values. The mean is represented by a white dot. Daily values are cumulative values of half-hourly values for evaporation and available energy, and daily mean for vapour pressure deficit.

Overall, annual evaporation followed the gradient in mean annual temperature, with evaporation increasing with increasing temperature (Figure 6). The highest annual evaporation was found at the Hisåsen site, with mean annual evaporation amounting




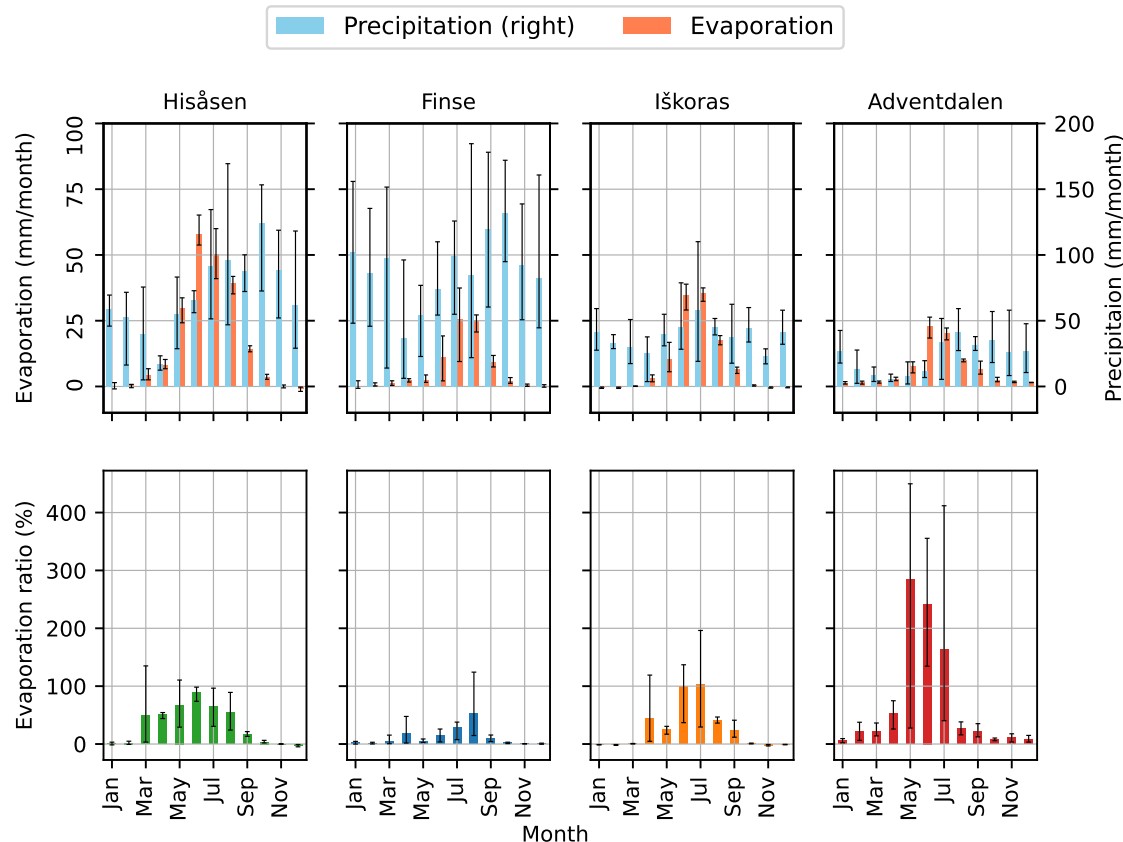

**Figure 5.** Mean monthly evaporation and precipitation in mm (upper row), with error bars representing minimum and maximum values, and mean monthly evaporation ratio in % (lower row), with error bars representing minimum and maximum values.

to 204 mm, followed by 107 mm at Iškoras, (81 mm) at Finse and (69 mm) at Adventdalen. Interannual variability in evaporation was generally low. At Hisåsen, Iškoras and Adventdalen, annual evaporation deviated less than 10 % from the mean. However, at Finse, the interannual variability was larger, with evaporation in 2020 being 34 % lower and in 2021 27 % higher than the mean of all four years.

We found that some site-specific interannual variability in evaporation corresponded with the interannual variability of the end of the snow-covered season. At Hisåsen, Finse and Iškoras the year with the lowest annual evaporation corresponded to the year with the longest lasting snow cover in spring (Fig. 6). Accordingly, at Hisåsen and Finse, the year with the highest annual evaporation corresponded to the year with the earliest snow cover melt-out. Comparing across sites, a larger interannual variability in snow-cover duration in spring corresponded with a larger interannual variability in evaporation, with Finse showing

the highest interannual variation.

The mean annual evaporation at our sites (81.1 mm to 207.6 mm), was in the lower range of the mean annual evaporation at the selected FLUXNET2015 sites (44.6 mm to 384.8 mm), despite being in the mid-range of mean annual temperature. Further-



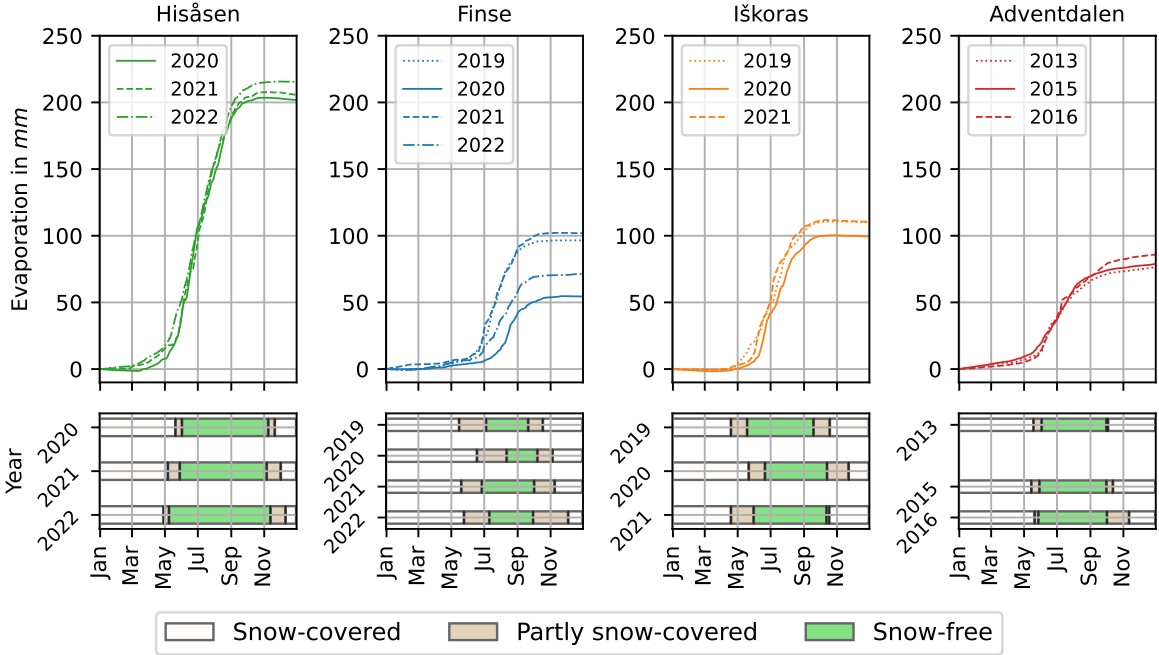

**Figure 6.** The upper row shows cumulative evaporation in mm (upper row) for the years 2019 (dotted lines), 2020 (dashed lines) and 2021 (solid lines) for sites Hisåsen (green), Finse (blue) and Iškoras (orange), and the years 2013 (dotted lines), 2015 (solid lines) and 2016 (dashed lines) for Adventdalen (red). The bars in the lower row represents the time of the year when the ground is either snow-covered (white bar) partly snow-covered (beige bar) or snow-free (green bar).

more, we found no significant linear relationship between annual evaporation with mean annual temperature when including the 14 FLUXNET2015 sites (Supporting figure C14). However, we found a significant increase in the annual evaporation with

mean temperature in the warm season (Figure 7), with a slope of 16.4 mm $°C^{-1}$ ($p<0.05$). Still, all of our sites were below the trendline. We were not able to detect any pattern in the deviation from the regression line based on ecosystem type, however, the ecosystem types were not evenly distributed over the temperature range. For example, all the forest sites were in the upper end of the temperature range.

## 4    Discussion

**4.1    Controls on evaporation**

Our results show that evaporation from northern latitude wetlands is mainly controlled by atmospheric evaporative demand, and furthermore, that the evaporative demand depends mainly on incoming solar radiation and vapour pressure deficit. On the sub-daily (hourly) timescale, vapour pressure deficit and incoming solar radiation had a strong correlation with evaporation, loaded





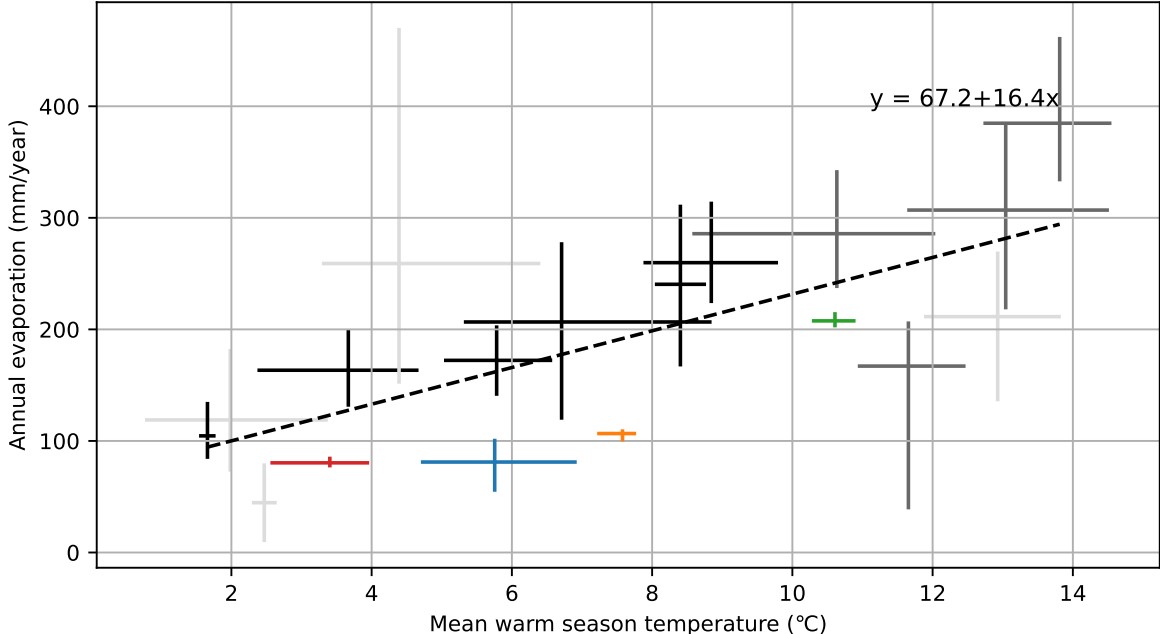

**Figure 7.** Annual evaporation of selected FLUXNET2015 sites above 60 °N latitude (evergreen needle-leaf forest in dark grey, wetlands in black and other ecosystem types in light grey) compared to the study sites Hisåsen (green), Finse (blue), Iškoras (orange) and Adventdalen (red). The annual evaporation (in mm) on the y-axis is plotted against mean May-September temperature (in °C), averaged over measured years, on the x-axis. The bars represent minimum and maximum values of years in measurement periods, while the intersect represent the mean. The dashed line shows the linear regression line of mean warm season (May-September) temperature and annual evaporation.

high in the factor analysis on the same factors as evaporation, and had a high relative importance in the random forest model.
Though vapour pressure deficit and incoming solar radiation had a high degree of shared variance, we found that evaporation was occasionally limited by low vapour pressure deficit despite available energy being high (Figure 3). Especially at Finse and Adventdalen, where the oceanic climatic influence lead to lower warm season temperatures and higher air humidity, the vapour pressure deficit was typically low, leading to constrained evaporation. At low vapour pressure deficit, more of the available energy was partitioned into sensible heat flux, as shown by the higher Bowen ratio for low vapour pressure deficit (Supporting
figure C8).

Other studies such as Liljedahl et al. (2011) and Helbig et al. (2020) have discussed the role of vapour pressure deficit in controlling northern latitude evaporation. Liljedahl et al. (2011) found latent heat flux to persistently exceed sensible heat flux when vapour pressure deficit was above 0.3 kPa for wet soils, and above 1.2 kPa for dry soils. We found a similar threshold at around 0.8 kPa to 1.0 kPa (Supporting figure C8) with no clear effect of soil moisture content (Supporting figure C9).
Additionally, we found that the Bowen ratio continued to decrease with vapour pressure deficit at Iškoras and Finse while the midday mean Bowen ratio at Hisåsen levelled off at values just below 1 at high vapour pressure deficits. The higher Bowen





ratio at Hisåsen during high vapour pressure deficit, might be caused by a higher cover of vascular plants. Vascular plants close their stomata during periods of high vapour pressure deficit to prevent excessive water loss (Novick et al., 2016), resulting in reduced transpiration rates. At Finse, Iškoras and Adventdalen, the footprints of the eddy covariance measurements have a

higher percentage than Hisåsen of open water surfaces and non-vascular vegetation where the soil remains saturated most of the year. Due to the low cover of vascular vegetation with stomatal control, evaporation will continue to increase in response to the increased atmospheric evaporative demand as vapour pressure deficit increases. Helbig et al. (2020) found a varying response to increasing evaporative demand based on vegetation type, and showed that evaporation from boreal peatlands can exceed that of boreal forest by up to 30 % during periods of high vapour pressure deficit above 2 kPa.

Our results indicate that evaporation during the snow-free season has a low sensitivity to surface conditions, apart from the effect of the surface on available energy. Overall, the evaporation dynamic in the snow-free season was well represented by the Penman-Monteith equation with a fixed site-specific surface conductance. We found no tendency for the relative error of the Penman-Monteith estimates or Bowen ratio to change with decreasing soil moisture content or time since rain (Supporting figure C5, C6 and C9), indicating that evaporation has a low sensitivity to soil moisture content at our sites. Furthermore,

we found no trend in relative error with growing degree day, indicating that the seasonal vegetation development has limited influence on the total evaporation. However, the transpiration changes due to leaf phenology and soil moisture variations may be undetectable due to a larger soil and free water evaporation in the total evaporation measured. Other studies have found that vegetation type is an important predictor of high-latitude evaporation (Oehri et al., 2022). The low sensitivity to surface conditions at our sites agrees with the results of Kasurinen et al. (2014), who studied evaporation in 65 boreal and arctic

eddy covariance sites and found that the surface tightly controls latent heat flux in mature forest, but has less importance in ecosystems with shorter vegetation such as grassland, wetlands, and tundra. Previous studies have found contrasting results regarding the sensitivity of evaporation to soil moisture content in northern latitude ecosystems. Westermann et al. (2009) and Liljedahl et al. (2011) found that the Bowen ratio in a high-arctic tundra site and arctic coastal wetland, respectively, decreased with higher soil water content, i.e., latent heat flux was favoured over sensible heat flux when soil water content was high. Ohta

et al. (2008) found soil moisture to be a strong control on interannual variation in evaporation from a deciduous needle leaf forest, while Sabater et al. (2020) found that temporal changes in soil moisture did not affect evaporative fluxes in subarctic deciduous woodland. Transpiration and soil evaporation typically decrease with soil moisture content below a certain threshold (Shutov et al., 2006). The four sites in this study can all be described as generally having well-watered soils throughout the year, and the measurement periods did not include substantial drought periods. Therefore, it is likely that the soil moisture

content did not reach a level where it would impact evaporation in the measurement periods.

We found that the presence or absence of snow on the surface had a large effect on evaporation rates, especially during spring, when the atmospheric evaporative demand was high. The amount of accumulated snow during the winter can affect the warm season flux budget. Pirk et al. (2023b) investigated the effect of snow cover duration on annual evaporation, in a study of water and carbon fluxes at Finse. In 2020, a year with one month delay in snow cover melt-out, the annual evaporation was

reduced by 50 % compared to 2021 – a year with normal snow cover duration. Stiegler et al. (2016) found that, in a year with above average snow accumulation, accumulated growing season latent heat flux was reduced by 33 % in a high arctic wet fen





in Zackenberg, Northeast Greenland, while it increased by 24 % in a nearby dry heath at the expense of sensible heat. We found that that sites with lower interannual variation in snow-cover duration have lower interannual variation in evaporation, and that a longer lasting snow-cover is typically associated with lower annual evaporation.

The dynamics of evaporation in the snow-free season was well represented by the Penman-Monteith equation, despite using a fixed surface conductance parameter. Predicting the magnitude, however, depends on finding a suitable value for surface conductance. The Penman-Monteith equation assumes surface energy balance, i.e. available energy is partitioned into either sensible or latent heat flux, whereas the measured fluxes do not amount to the measured available energy at the sites. Accordingly, the surface conductance parameter acts to compensate for the lack of for energy balance closure. By optimizing

the surface conductance parameter using the sum of observed sensible and latent heat as available energy (i.e. forcing energy balance closure), we found higher values of surface conductance, with less variation between sites ($0.0028$ ms$^{-1}$ to $0.0044$ ms$^{-1}$ compared to $0.000$ ms$^{-1}$ to $0.0028$ ms$^{-1}$ when the difference between net radiation and soil heat flux was used as available energy). Lack of energy balance closure is not uncommon for eddy covariance sites, and can be caused by e.g. unmeasured storage terms, large scale exchange processes, and landscape heterogeneity (Foken, 2008; Stoy et al., 2013).

## 4.2    Magnitudes of evaporation

At the four Norwegian eddy covariance sites studied, we found that the mean annual evaporation ranges between 80 mm and 208 mm. This is at the lower end and below the range of pan evaporation rates measured at 42 sites in Norway between 1967 and 1972 Hetager and Lystad (1974). The pan evaporation sites were mostly located at lower altitude with warmer temperature. For the 5 pan sites with available mean annual temperature in the reference period 1961-1990 (based on data from MET Norway)

below 3 °C, the annual pan evaporation ranged from 250 mm to 315 mm in a range of mean annual temperature from 1.0 °C to 2.4 °C.

The magnitudes of annual evaporation at our four Norwegian eddy covariance sites are within the range found at the northern latitude eddy covariance sites in FLUXNET2015 (44.6 mm to 384.8 mm). Including the FLUXNET2015 sites, we found no significant linear relation of annual evaporation to spatial variations in annual mean temperature, but a strong relation to mean

temperature in the warm season (May-September). This can be expected, since most of the annual evaporation happens in the warm season. Furthermore, the annual temperature is sensitive to effects of continentality, such as very cold winters, that don't influence annual evaporation. As sites in continental climate typically will have cold winters and warm summers, while sites in oceanic climates typically have milder winters and cooler summers, the annual temperature is not necessarily a good proxy for warm season temperature. A high warm season temperature increases evaporation through increasing the air's ability to hold

water and thus increasing the vapour pressure deficit. High air temperature is also typically linked to high available energy, which again is an important control on evaporation. When considering the dependence of the annual evaporation on warm season temperature, all the Norwegian sites ended up below the regression line. Grouping the site ecosystem type into wetland and forest sites did not reveal any pattern that could explain the deviations from the trend line. However, the ecosystem types were not evenly distributed in the temperature range. One possible explanation for the deviations is the degree of continentality,





where sites in more oceanic climates are influenced by maritime wet air and have lower vapour pressure deficit for the same temperatures compared to sites in more continental climates.

We found a large variation in the role that the evaporation plays in the local water balance. At Finse, the annual evaporation was less than 13 % of the annual precipitation, while at Adventdalen evaporation amounted to more than 28 % of the precipitation. The large variation in the evaporation ratio was caused both by a low variation in annual evaporation across sites compared

to precipitation, and the fact that the wettest site, Finse, is also the site with the lowest evaporation. As the precipitation data used in this study have not been corrected for potential under-catch, the precipitation might be underestimated, especially in the months when precipitation falls as snow. Thus, the estimated evaporation ratios might be overestimated, especially estimates for months in the cold season and annual scale estimates. The estimates of evaporation ratio for months in the warm season are less likely to be affected by the error from precipitation under-catch, as under-catch is not a big problem when precipitation

falls as rain (Wolff et al., 2015).

### 4.3    Northern latitude wetland evaporation in a warmer climate

In a warmer climate, the snow-cover duration is expected to decrease (Rizzi et al., 2017). It is also expected that vapour pressure deficit will increase (Novick et al., 2024), due to a higher expected increase in saturated vapour pressure with temperature relative to the increase in actual vapour pressure. Both the earlier start of the snow-free season, the increased vapour pressure

deficit, and the expected tree line expansion could enhance evaporation in northern latitudes (Nicholls and Carey, 2021). The timing of the snow cover melt-out can affect the evaporative demand by affecting the energy available for evaporation. If the snow cover melts out earlier, less solar radiation will reflect due to the lower albedo of a snow-free compared to a snow-covered surface. Furthermore, less energy will be allocated for snow melt. The sites in this study had mean snow melt-out dates from 23.05 to 14.07, a period of the year when incoming solar radiation typically is high. Water availability is also typically high due

to the recharge of soil moisture from snow-melt. The combination of earlier snow-free ground and increased vapour pressure deficit, in a season when the incoming solar radiation and water availability is high, will likely increase evaporation in spring. The timing of the start of the snow covered season in autumn is less likely to affect the evaporation, as incoming solar radiation is typically low in this period of the year.

Our results show that low vapour pressure deficit is a constraining factor for evaporation in the northern latitudes, especially

in coastal regions. It is likely that evaporation will increase with the expected increased vapour pressure deficit. The response of the ecosystem evaporation to increased vapour pressure deficit will, however, depend on soil moisture availability. Though we did not observe any constrain on evaporation from low soil water availability, evaporation was regularly exceeding precipitation in May, June, and July at the site with the lowest precipitation and was typically around 50 % of precipitation in the warm season. The soil moisture availability therefore depends on precipitation and/or melt water from preceding months. An earlier

snow-cover melt-out and increased evaporation in spring, might lead to lowered soil moisture availability later in the season. Furthermore, the response of ecosystem evaporation to increased vapour pressure deficit will depend on the response of the vegetation to the increased evaporative demand. Vascular plants can limit transpiration to avoid excessive water loss when the evaporative demand is high (Novick et al., 2016).



## 5   Conclusions

We investigated controls and magnitudes of evaporation from four northern latitude wetlands in Norway. Our analysis show that the hourly evaporation in the snow-free season is mainly controlled by the atmospheric evaporative demand, which again is mainly controlled by incoming solar radiation and vapour pressure deficit. Our results indicate that the evaporation had a low sensitivity to the phenology and observed changes in soil water content. We found that the mean daily evaporation was highest in June–August at all sites. However, the timing of the highest mean daily rates were slightly different between the

sites. Mean annual evaporation ranged from 81 mm to 208 mm and increased with the spatial gradient in warm season mean temperature. We found that that sites with lower interannual variation in snow-cover duration have lower interannual variation in evaporation, and that a longer lasting snow-cover is typically associated with lower annual evaporation. The magnitudes of annual evaporation at our four Norwegian eddy covariance sites were within the range found at the northern latitude eddy covariance sites in FLUXNET2015, but in the lower range when considering the spatial gradient in the warm season mean

temperature.

The variability in ET found across our sites underpins the pressing need for additional in-situ measurements. These data-scarce regions are projected to experience strong climate warming, which can feed back to other components in the Earth system.

*Code availability.*   Python scripts for analysis and plotting are available at https://github.com/pasta4u/north_lat_evap.

*Data availability.*   The gap filled time series of evaporation and local meteorological variables are available at https://doi.org/10.5281/zenodo.10044324.

*Author contributions.*   Conceptualization: AV, KE, NP, LMT, AVV; Data curation: AV, NP; Formal analysis: AV; Funding acquisition: LMT, NP; Writing - original draft preparation: AV; Writing – review and editing: AV, KE, NP, LMT, AVV

*Competing interests.*   The authors declare that they have no conflict of interest.

*Acknowledgements.*   This work is a contribution to the strategic research initiative LATICE (Faculty of Mathematics and Natural Sciences, University of Oslo, Project #UiO/GEO103920). We thank Anders Bryn and Peter Horvath at the Natural History Museum, University of Oslo, Norway, for kindly sharing their findings after performing vegetation mapping analyses in the footprints of the three mainland towers.





We thank Poul Larsen and his team from DMR for operating the Hisåsen site. This work was supported by the Research Council of Norway (project #301552 (Spot-On)) and the European Research Council (project #101116083 (ACTIVATE)).



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





**Appendix A: Climate reference data**

**Table A1.** Coordinates, altitude, mean annual precipitation (MAP) and temperature (MAAT) for weather stations used to describe the climatic context of the sites in this study. MAP and MAAT are averages over the climate reference period 1991-2020. Based on data from The Norwegian Meteorological Institute.

| Station ID | Station name | Coordinates | Altitude (m.a.s.l) | MAAT (°C) | MAP (mm) |
|---|---|---|---|---|---|
| SN25830 | Finsevatn | 60.59N 7.53E | 2010 | 1.1 | 968 |
| SN180 | Trysil Vegstasjon | 61.29N 12.27E | 360 | 2.7 | 857 |
| SN97710 | Iskoras II | 69.30N 25.34E | 591 | -1.6 | |
| SN97251 | Karasjok - Markannjarga | 69.46N 25.50E | 131 | -1.2 | 417 |
| SN99840 | Svalbard lufthavn | 78.24N 15.50E | 28 | -3.9 | 218 |

645    Mean annual precipitation (MAP) and temperature (MAAT) listed in Table A1 are averages over the climate reference period 1991-2020 from the nearest MET Norway weather station. Data from the nearest weather station is used to estimate MAP and MAAT values for each site (Table 1) and to compare the monthly temperature and precipitation in the measurement periods to that of the climate reference period (Figure 2). The nearest weather station to Hisåsen is "SN210 - Trysil Vegetasjon" (357 m.a.s.l, precipitation and temperature). For Finse, the weather station is collocated with the eddy covariance station and climate

650    reference data listed is from this station. For Iškoras, the nearest weather stations are "SN97251 Karasjok - Markannjarga" (131 m.a.s.l, precipitation and temperature), and "SN97710 Iskoras" (131 m.a.s.l, temperature only). An altitude-based weighted average between the two nearest weather stations is used to determine the mean annual and monthly values for Iškoras (380 m.a.s.l.) in Figure and Table 1.



# Appendix B: List of FLUXNET2015 sites used for comparison

**Table B1.** Sites in FLUXNET2015 selected for comparison. Selected sites are located at latitude above 60 °N and have data licence CC-BY-4.0.

| SITE ID | SITE NAME | LATITUDE | LONGITUDE | ELEVATION | IGBP | MAAT | MAP |
|---------|-----------|----------|-----------|-----------|------|------|-----|
| FI-Hyy | Hyytiala | 61.8474 | 24.2948 | 181 | ENF | 3.8 | 709 |
| FI-Jok | Jokioinen | 60.8986 | 23.5134 | 109 | CRO | 4.6 | 627 |
| FI-Let | Lettosuo | 60.6418 | 23.9595 | 111 | ENF | 4.6 | 627 |
| FI-Lom | Lompolojankka | 67.9972 | 24.2092 | 274 | WET | -1.4 | 484 |
| FI-Sod | Sodankyla | 67.3624 | 26.6386 | 180 | ENF | -1 | 500 |
| GL-NuF | Nuuk Fen | 64.1308 | -51.3861 | 50 | WET | -1.4 | 750 |
| GL-ZaF | Zackenberg Fen | 74.4814 | -20.5545 | 38 | WET | -9 | 211 |
| GL-ZaH | Zackenberg Heath | 74.4733 | -20.5503 | 38 | GRA | -9 | 211 |
| RU-Che | Cherski | 68.613 | 161.3414 | 6 | WET | -11 | 197 |
| RU-Cok | Chokurdakh | 70.8291 | 147.4943 | 48 | OSH | -14.3 | 232 |
| SJ-Blv | Bayelva, Spitsbergen | 78.9216 | 11.8311 | 25 | SNO | -4.5 | 400 |
| US-Atq | Atqasuk | 70.4696 | -157.4089 | 15 | WET | -9.7 | 93 |
| US-Ivo | Ivotuk | 68.4865 | -155.7503 | 568 | WET | -8.28 | 304 |
| US-Prr | Poker Flat Research Range Black Spruce Forest | 65.1237 | -147.4876 | 210 | ENF | -2 | 275 |



655 **Appendix C: Supporting figures**

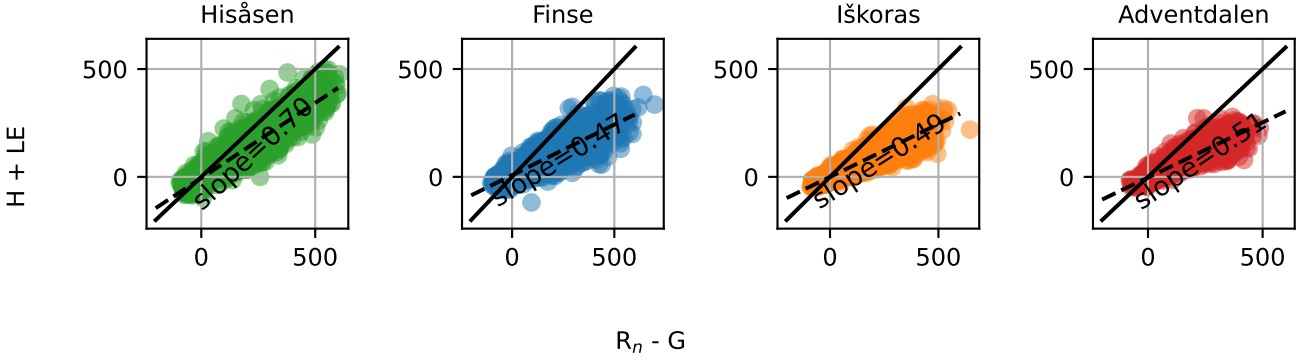

**Figure C1.** Energy balance closure in the snow-free season for Hisåsen (green), Finse (blue), Iškoras (orange) and Adventdalen (red). The energy balance is estimated from the slope of the linear regression of the sum of latent and sensible heat on the x-axis against available energy (difference between net radiation and soil heat flux) on the y-axis.

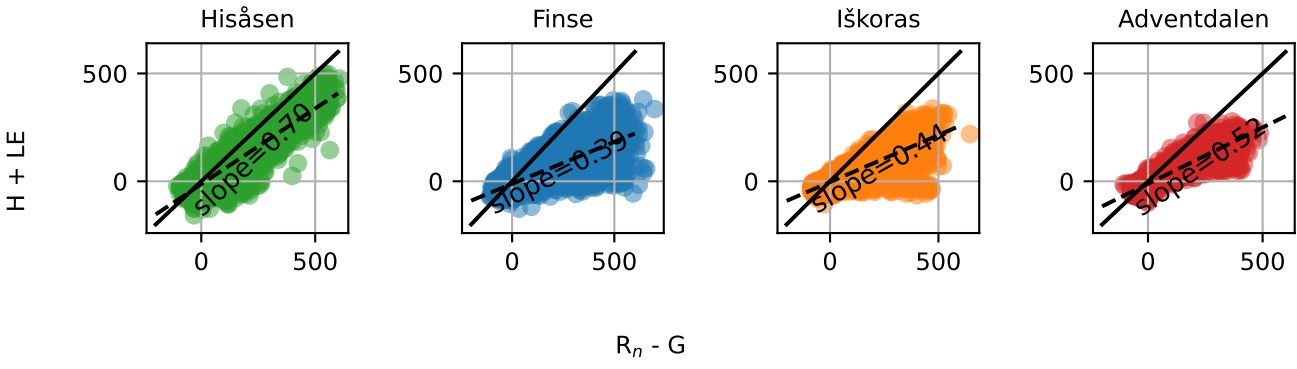

**Figure C2.** Energy balance closure (whole year) for Hisåsen (green), Finse (blue), Iškoras (orange) and Adventdalen (red). The energy balance is estimated from the slope of the linear regression of the sum of latent and sensible heat on the x-axis against available energy (difference between net radiation and soil heat flux) on the y-axis.

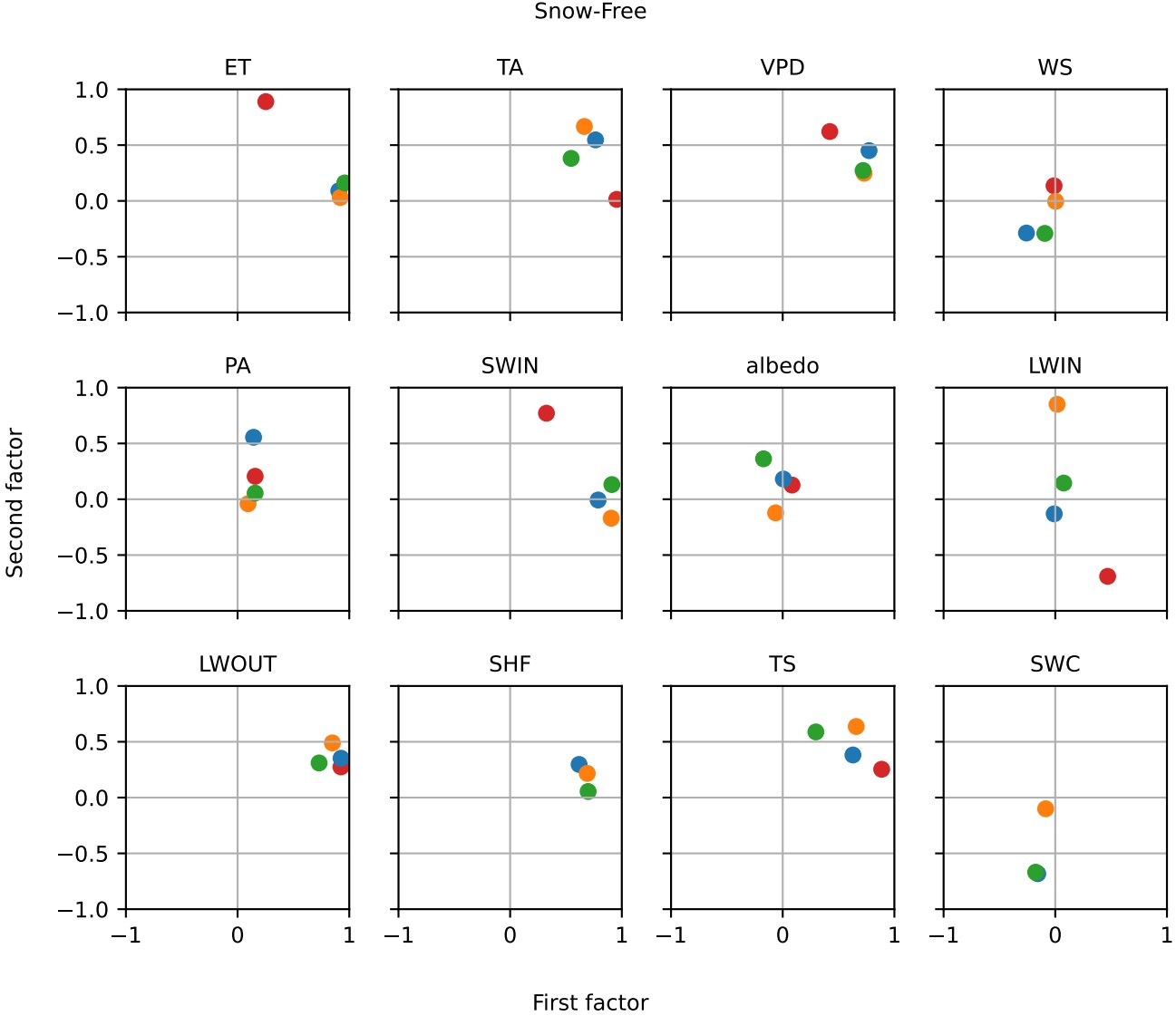

**Figure C3.** Factor loadings of first and second factor in the snow-free season, for each of the variables in the dataset. In each subplot the points represent results from Hisåsen (green), Finse (blue), Iškoras (orange) and Adventdalen (red).



**Figure C4.** Factor loadings of first and second factor in the snow-covered season, for each of the variables in the dataset. In each subplot the points represent results from Hisåsen (green), Finse (blue), Iškoras (orange) and Adventdalen (red).



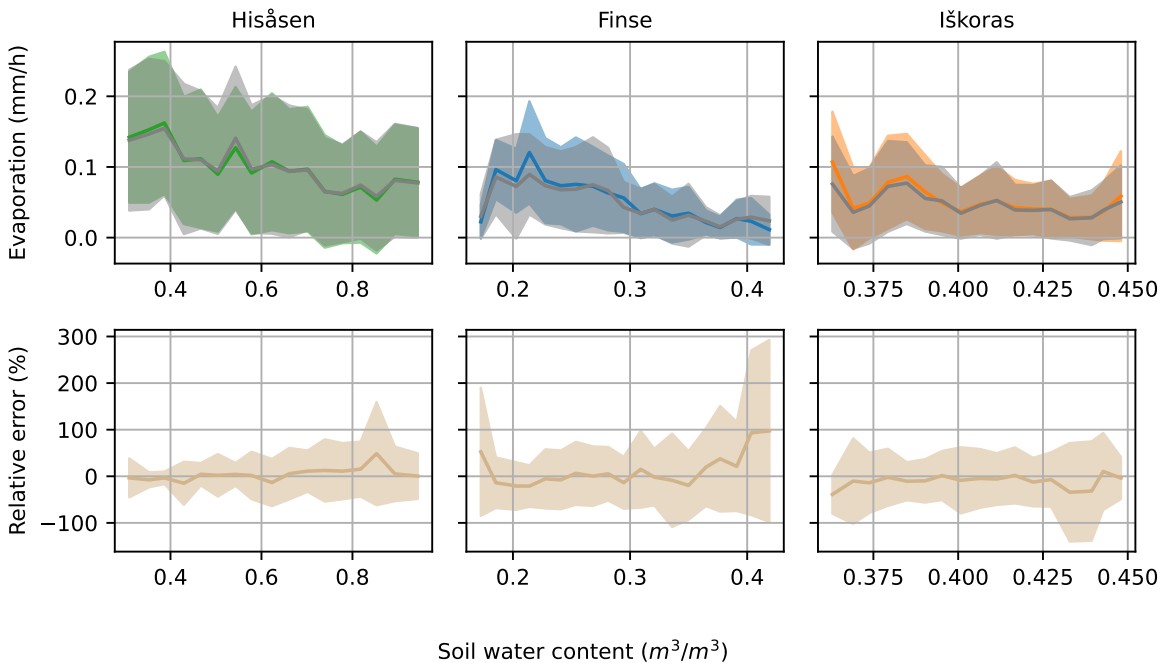

**Figure C5.** Sensitivity of observed (coloured) and modelled (grey) evaporation (upper row) and the relative error (lower row) to volumetric soil water content.



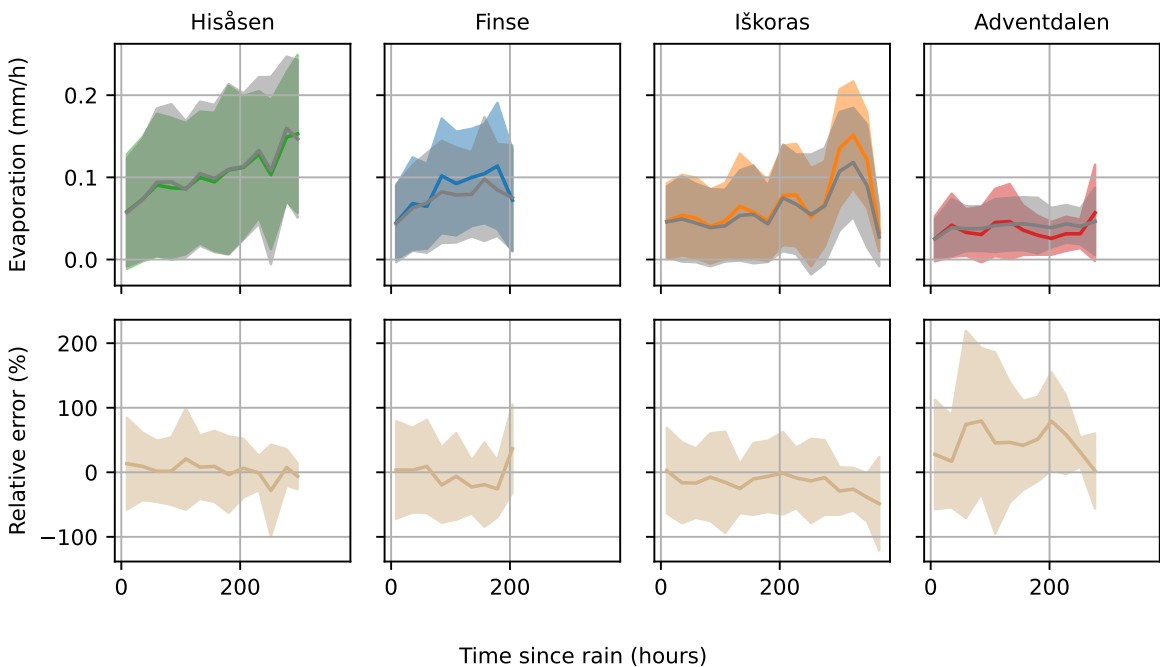

**Figure C6.** Sensitivity of observed (coloured) and modelled (grey) evaporation (upper row) and the relative error (lower row) to time since rain.





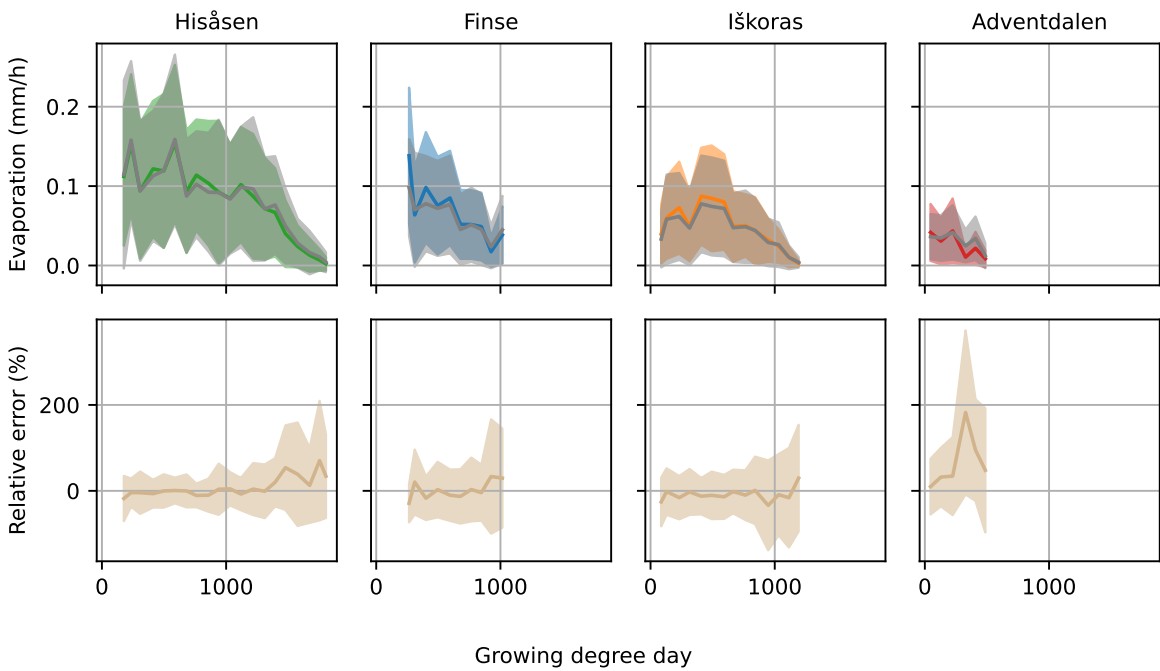

**Figure C7.** Sensitivity of observed (coloured) and modelled (grey) evaporation (upper row) and the relative error (lower row) to growing degree day.

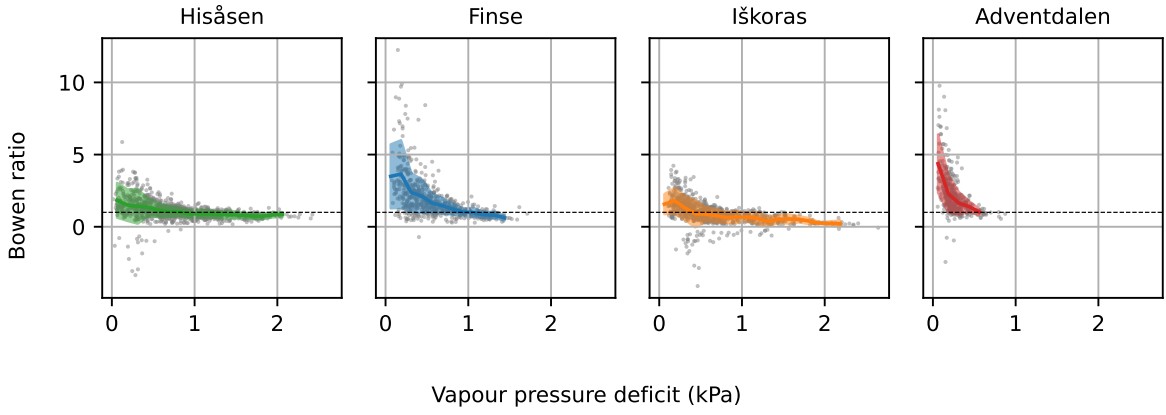

**Figure C8.** Sensitivity of Bowen ratio to vapour pressure deficit. The coloured areas show the mean (+/- standard deviation) midday Bowen ratio (grey dots) for 20 bins of vapour pressure deficit.



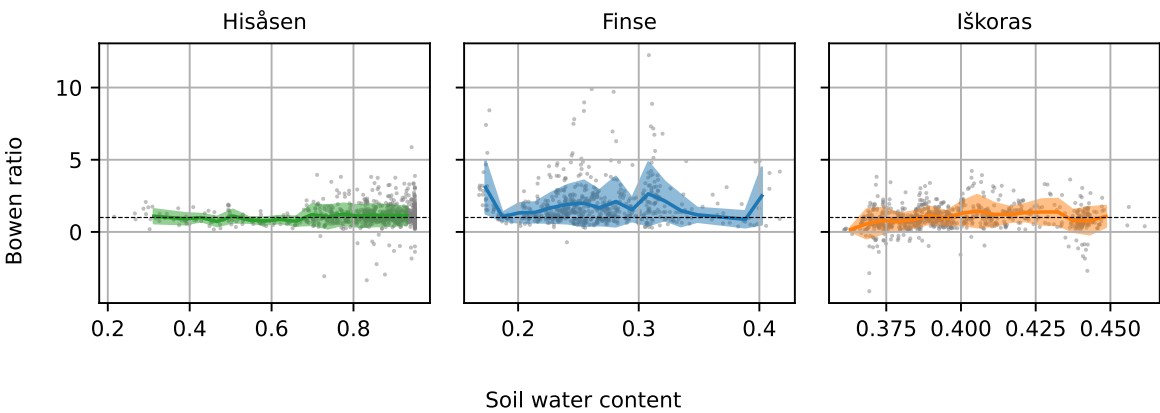

**Figure C9.** Sensitivity of Bowen ratio to volumetric soil moisture content. The coloured areas show the mean (+/- standard deviation) midday Bowen ratio (grey dots) for 20 bins of volumetric soil moisture content.





**Figure C10.** Distribution of daily evaporation (upper row, in mm d$^{-1}$), net radiation (middle row, in MJ d$^{-1}$m$^{-2}$) and vapour pressure deficit (lower row, in kPa) at Hisåsen. For each month, the box plot represents median, 25- and 75 quantiles, and whiskers represent minimum and maximum values. The mean is represented by a white dot. Daily values are cumulations for evaporation and net radiation, and daily mean for vapour pressure deficit.



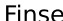

**Figure C11.** Distribution of daily evaporation (upper row, in mm d$^{-1}$), net radiation (middle row, in MJ d$^{-1}$m$^{-2}$) and vapour pressure deficit (lower row, in kPa) at Finse. For each month, the box plot represents median, 25- and 75 quantiles, and whiskers represent minimum and maximum values. The mean is represented by a white dot. Daily values are cumulations for evaporation and net radiation, and daily mean for vapour pressure deficit.





**Figure C12.** Distribution of daily evaporation (upper row, in mm d$^{-1}$), net radiation (middle row, in MJ d$^{-1}$m$^{-2}$ and vapour pressure deficit (lower row, in kPa) at Iškoras. For each month, the box plot represents median, 25- and 75 quantiles, and whiskers represent minimum and maximum values. The mean is represented by a white dot. Daily values are cumulations for evaporation and net radiation, and daily mean for vapour pressure deficit.





Adventdalen

**Figure C13.** Distribution of daily evaporation (upper row, mm d$^{-1}$), net radiation (middle row, in MJ d$^{-1}$m$^{-2}$) and vapour pressure deficit (lower row, in kPa) at Adventdalen. For each month, the box plot represents median, 25- and 75 quantiles, and whiskers represent minimum and maximum values. The mean is represented by a white dot. Daily values are cumulations for evaporation and net radiation, and daily mean for vapour pressure deficit.



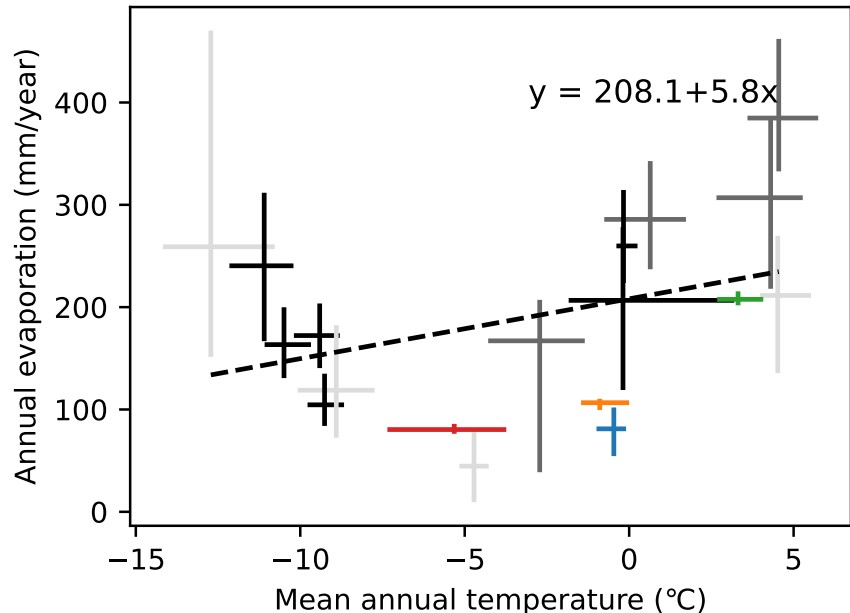

**Figure C14.** Annual evaporation of selected FLUXNET2015 sites above 60 °N latitude (evergreen needle-leaf forest in dark grey, wetlands in black and other ecosystem types in light grey) compared to the study sites Hisåsen (green), Finse (blue), Iškoras (orange) and Adventdalen (red). The annual evaporation (in mm) on the y-axis is plotted against annual mean temperature (in °C), averaged over measured years, on the x-axis. The bars represent minimum and maximum values of years in measurement periods, while the intersect represent the mean. The dashed line shows the linear regression line of annual mean temperature and annual evaporation. The regression slope was not significant at $p>0.05$.