# Peer review of "Evaporation from northern latitude wetlands"

_EGUsphere, 2025_

## Referee Comment (RC2)

**Evaporation from northern latitude wetlands**

Astrid Vatne1, Norbert Pirk1, Kolbjørn Engeland1,2, Ane V. Vollsnes3, and Lena M. Tallaksen1

1Department of Geosciences, University of Oslo

2Norwegian Water Resources and Energy Directorate

[referee-annotated manuscript omitted]

---

## Author Comment (AC1)

**Response to comment by Anonymous Referee #1**

Thank you very much for the thorough and constructive review. Hereby, we would like to respond to your comments (in the following, the comments from the reviewer (AR#1) are in plain text, and our responses are in bold text).

**General Comments**

AR#1: The authors present multiple years of eddy covariance data from four northern sites across Norway and compare annual evapotranspiration totals and evaporative controls to other northern FLUXNET sites. The presented data is a valuable contribution to the field due to the scarcity of evaporation measurements at northern sites, particularly over both the snow-covered and snow-free seasons, and the authors present the interannual variability of evaporative totals and controls across their sites. It is a lot of work to compile eddy covariance data from that many site-years and I commend the authors for their efforts, presentation, and gap-filling techniques. I appreciate how well they have contextualized their sites within current (albeit fairly scarce) literature and evaporation data from other northern sites.

**Response: Thank you very much for this positive comment.**

AR#1: Although the writing and figures are clear, I find the abstract, discussion (mainly sections 4.2 and 4.3) and main conclusions do not entirely explain the results of the study in enough detail as the paper is currently written. This comment may be a compliment to the paper, as there were interesting results and nuances among the sites that were presented in the results, yet not fully explained in the discussion. The authors often repeat themselves, explaining that a warmer climate and increased atmospheric demand for moisture will increase ET (which is already fairly well established), yet do not fully dive into the interesting results and differences between sites. The authors reiterate that a longer snow-free period will increase overall evaporation and that their sites had low sensitivity to the phenology and soil water content. While these are important conclusions, it also does appear that there are variations among these

four sites presented that could be explored in more detail with even more added scientific value (i.e. different threshold responses to VPD, varying sensitivities to soil moisture, etc.). Additionally, the Penman vs. observed ET at these sites is a useful comparison to know how to deal with surface conductance terms in the future.

**Response: We agree and will update the abstract, discussion and main conclusions to focus more on the differences in evaporation controls between the sites. More specifically, we will discuss the differences between sites in how they correlate with controlling variables such as air temperature and vapour pressure deficit, and the differences in Bowen ratio sensitivity to vapour pressure deficit. We will discuss the observed differences in light of the site characteristics such as vegetation structure and presence of permafrost.**

AR#1: Overall, this paper is a useful and well-presented contribution to furthering our understanding year-round northern wetland evaporation. However, the authors may want to consider reframing the primary messaging of the paper away from describing the ET totals within the context of the water balance (which is tricky when only comparing ET and precipitation and not even presenting snow accumulation data) to focusing on highlighting the different controls/sensitivities between these sites, even if subtle.

**Response: Thank you for your positive comment. We agree that the focus should be revised to include a discussion on differences in evaporation controls between the sites. As we have a hydrological perspective and the vertical water balance is important for downstream areas, we propose to keep the evaluation of ET totals in context of the water balance. We will however use the term "vertical water balance" where relevant, as you suggested in the technical corrections. As for snow accumulation, we chose not to include such data, firstly, because we do not have good snow data available (particularly for snow water equivalent, SWE, observations). Secondly, as our focus is on fluxes and not storage, we argue that including only the presence/absence of snow cover is sufficient for our discussion of evaporation totals in the context of the water balance. We will however, in the revised manuscript, include model estimates of SWE and how much of the precipitation that falls as snow at the different sites.**

**Title**

AR#1: I appreciate the simplicity of the title and understand if the authors choose to keep it as-is. I do, however, think there may be more value in adding

in key words highlighting that the paper evaluates evaporative controls across these sites, has year-round measurements, and includes tundra ecosystems. It could also mention the context of Norway, etc. It currently sounds as though this is a review paper rather than a paper that presents new data.

**Response: We agree that the title could be more specific and suggest changing the title to "Year-round measurements of evaporation from northern latitude wetlands in Norway".**

**Abstract**

AR#1: While the paper is well written, the abstract does not capture the interesting findings that were presented throughout the paper. The abstract could emphasize both the analysis that was performed, and the dataset a bit more, particularly the fact that these are year-round measurements and the shoulder season processes are of value with variable snow-cover. I believe this paper does improve our understanding of evaporation from these sparsely studied ecosystems, but as it is written, the abstract does not really present any new findings or insights (that again, the paper results do). I do not believe the current abstract does the paper justice, and I worry those skimming it would miss the valuable contributions this work does present.

**Response: Thank you for this valuable remark. We agree and will add more information on the analysis and dataset. Further, we will ensure to better reflect on the observed differences between the sites.**

**Introduction**

AR#1: The introduction needs to discuss the importance and extent of wetlands in this region more specifically. There is discussion of thaw lake basins, alpine tundra sites, etc. and the authors state that most other studies focus on forested areas and carbon, however the justification behind why we need wetland-focused studies is lacking. It would be useful to know their proportion of northern landscapes, their hydrological role in larger ecosystem functioning, unknown sensitivities to climate change, etc. The authors state these ecosystems are largely missing from the literature, but justification behind why we need this information is not currently presented.

**Response: We agree and will add a paragraph discussing the extent and importance of wetlands in the region, as documented by references such as Bacon et al. (2017) Mires and Peat, 19, 12., Bryn et al. (2018) Norwegian Journal of Geography, 72(3), 131–145., Xu et al. (2018) CATENA, 160, 134-140.**

AR#1: Additionally, it would be useful for the introduction to briefly discuss some of the processes/controls on evaporation in a bit more detail. For example, in the discussion you reference Liljedahl et al. (2011) and Helbig et al. (2020) and discuss VPD controls on ET and bowen ratios with varying soil moisture and the differences between snow-covered and snow-free seasons. This of course can remain in the discussion; however, some mention of these processes would be useful upfront in the introduction.

**Response: We agree and will include a more detailed introduction to the key processes controlling evaporation in northern ecosystems, referring to studies such as Brümmer et al. 2012. Agricultural and Forest Meteorology 153: 14-30, Liljedahl et al. 2011. Biogeosciences 8: 3375-3389, Thunberg et al. 2021. Atmosphere 12: 1359, and Helbig et al. 2020. Nature Climate Change 10: 555-560.**

**Methods**

AR#1: I understand why the authors use time since rain as a proxy for soil moisture availability and I do think this is fine for their presented analysis, however it would be useful to know how their time since rain variable compared to the point measurements they also took. This warrants a bit more discussion or explanation (even one or two sentences), as precipitation alone of course does not describe soil moisture dynamics. There is a large difference in soil moisture if it rained 0.1 mm (their threshold for a rain event) vs. 20 mm over a short period. Particularly without the context of vegetation structure or soil properties.

**Response: We use time since rain as a proxy of surface water availability, primarily for use in the gap filling routine, since point measurements of soil moisture is not available at all sites. Further, because time since rain could represent surface water that is not captured by the soil moisture sensors, such as intercepted water. We chose a threshold of 0.1 mm/h since this is the limit of a detectable rain event. We agree that rain intensity and duration does make a difference, though we would argue that since the magnitude of our fluxes are low, 0.1 mm is a relevant amount to consider. We agree that a comparison between time since rain and point measurements of soil moisture would be useful, and will include it in the revised manuscript. Further, we will include a general discussion of soil moisture availability at each site.**

AR#1: Additionally, I initially assumed the authors would discuss changing precipitation patterns with climate change as rationale for using time since rain – are there sensitivities to drought or more frequent precipitation? This is not discussed. Is there any further information on soil properties?

**Response: Time since rain was included primarily as a proxy for surface water availability in the gap filling routine. It was not our intention to look at changes in time since rain or other surface variable, although this would be a very interesting aspect to include in a future study. Unfortunately, we do not have a systematic characterization of soil properties that is consistent for all sites.**

AR#1: Is it reasonable to use time since rain = 0 during the snow-covered season?

**Response: As we intended the time since rain variable as a surface water proxy (mainly for use in the gap filling routine), we would argue that it is reasonable to use time since rain = 0 during the snow-covered season. Although it is in a frozen state, water is readily available for evaporation when the ground is snow-covered. Though the evaporation process is not equal for liquid water (evaporation) and snow (sublimation and evaporation of melted snow), our idea was that the difference in evaporation process for snow and water could be captured by the regression model (see section 2.2.1) using information from other predictors such as snow-cover status and temperature.**

AR#1: In my experience, evaporative fluxes measured with eddy covariance in the snow-covered and shoulder seasons are challenging in terms of poor data quality. It would be useful for the reader to have an idea of how much gap-filling was done per month of measurement once the QAQC on the data was performed.

**Response: We agree and will add a table showing the percentage of gap-filled data for each month.**

AR#1: In the site descriptions, you mention specific vegetation for some types and not others.

**Response: Thank you for pointing this out. We will update the site description to include typical vegetation at each site.**

AR#1: The energy balance closure is presented in the appendix but more thorough explanation of the differences between the snow-free and full year energy balance closures could improve the paper. It is discussed in relation to the surface conductance, but I feel it could use further explanation.

**Response: We agree and will include a paragraph discussing the difference between the snow-free and full year energy balance closures.**

**Results**

AR#1: Line 279: Is there a physical reason why there are higher relative errors at high values of soil water content at Finse and Hisaasen?

**Response: Reviewing the data and figures, we see that this is only the case at Finse. We will check what conditions are correlated with values of soil water content ¿ 0.4 $m^3/m^3$ at Finse, and why this is not the case at the other sites.**

AR#1: Line 320-324: The year with the lowest annual evaporation corresponded to the longest lasting snow cover – Can you discuss whether this was due to similar ET rates that just lasted longer/had more days to evaporate? Or perhaps the year with the earlier snowmelt had warmer/sunnier conditions or higher soil moisture? Higher VPD? Was the total ET higher simply because it was a longer season or were there other factors? Later in the discussion, the authors seem to indicate this could be due to lower albedo with no snow cover, however I think some of the above factors could also be investigated in a dataset like this.

**Response: Thank you for the suggestions. We will use ANOVA to test if daily evaporation, vapour pressure deficit and incoming- and net shortwave radiation in the snow-free season are significantly different between years.**

**Discussion**

AR#1: Line 346-359: You discuss the role of vegetation here and in the following paragraph, but I think this could be expanded a bit more to explain other results (i.e. perhaps why Finse has higher sensitivity of Bowen Ratio to

mid-range soil moistures? Fig C9. Is this vegetation influence?). The general messaging in the paragraph ending in Line 380 is that the sites are well-watered and vegetation plays a very minor role. This may be true, but I think more information on the vegetation structure of the sites (even photos) may help.

**Response: We agree and will add more information on the vegetation structure and its potential role in explaining the variability of the sites, including a figure with photos from the sites.**

AR#1: Line 389: You found sites with lower interannual variation in snow-cover duration also have lower interannual variation in evaporation, but can you comment on the amount of accumulated snow in these years? How is this related to the amount of snow? Of course, increased snow depth will also likely result in later snow cover, but is it simply the presence/absence of snow, or the amount of added moisture to the system? Likely just the presence, as these systems are not moisture-limited, but perhaps worth including some snow data if possible.

**Response: As commented earlier (in response to AR#1—General comments), we suggest not emphasizing the discussion on snow accumulation, due to the low availability of observations. We will, however, include model estimates of snow water equivalent for the sites. These estimates indicate that the snow-cover duration in spring is indeed controlled by the amount of accumulated snow. Further, we will include a general discussion of soil moisture availability at each site and assess whether the years with more accumulated snow affect the site soil moisture.**

AR#1: Line 401-421 discussing the magnitude of ET and influence of temperature: This is a lengthy paragraph to explain that warmer temperatures and humidity influence ET. I am not sure there is much added value here. I recommend that the discussion focus more on differences between the studied sites and physical processes that may be causing these, rather than explaining these general relationships that are already well understood.

**Response: We agree that the paragraph is too lengthy compared to the added scientific value. We do, however, think it is interesting to discuss the magnitude of evaporation at our sites to that of other sites in the region, and suggest keeping a shorter version of the paragraph in the revised manuscript. Further, we will elaborate the discussion of the differences between the studied sites, such as the sensitivity to vapour pressure deficit.**

AR#1: Line 422: Be careful with stating the "local water balance" as this is simply two components of the water balance

**Response: We agree and will rather use the term "vertical water balance", as you suggested in the technical corrections.**

AR#1: The last paragraph of section 4.2 does not read as a discussion.

**Response: We are unsure what part of the discussion the reviewer is referring to, and would argue that the last paragraph of section 4.2 does read as a discussion.**

AR#1: Line 446: References here too (i.e. Helbig et al., 2020, etc)

**Response: We agree and will include references such as Novick et al. 2016.Nature Climate Change 6: 1023-1027 .**

AR#1: Section 4.3: This section reads more like an introduction than a discussion. The first paragraph states what we already predict – a longer snow-free season will increase evaporation. These results are not adequately discussed here. The second paragraph discusses VPD as a constraining factor for evaporation, which is a point related to some interesting results you present for each site. However, you do not really discuss your sites or results specifically at all. It ends with discussion of vascular plants limiting transpiration, which you mention above likely is not a large control at your sites.

**Response: We agree and will carefully review and rewrite the section. The original idea of the section was to discuss our results in light of expected changes in evaporation controls. However, we realise that it reads more like an introduction.**

AR#1: Why was VPD the only significant control at Adventdalen in the snow-covered season? (Line 253?) It seems to be behaving differently and any physical justification for this is not adequately discussed.

**Response: Thank you for pointing this out. We suggest adding the following to the discussion on evaporation control: "We found that**

evaporation at Adventdalen in the snow-covered season was mainly controlled by vapour pressure deficit. The correlation to other controls was weak. This is likely due to the long polar night at Adventdalen with low variation in solar energy". Further, we will look into other factors such as variation in vapour pressure deficit due to the maritime setting".

**Additional specific comments**

AR#1: Line 20: Do you have any references for evaporation being a minor component of the annual water balance?

**Response: Yes, we will include references such as Erlandsen et al. 2021 Hydrology Research 52: 356–372. in the revised manuscript.**

AR#1: Line 24-27: Reference Penman, etc? There are older references here that summarized these processes before the references you listed.

**Response: We agree and will update the references to include Penman 1948. Proceedings of the Royal Society of London. A. Mathematical and Physical Sciences; 193 (1032): 120–145.**

AR#1: Line 27: Could use VPD instead of writing it out each time

**Response: Although "vapour pressure deficit" is a rather lengthy term, we prefer avoiding abbreviations in the text.**

AR#1: Line 67: boreal is not a Koppen classification – This needs to be fixed.

**Response: Thank you for pointing out this error. We will update the description of site climates.**

AR#1: Line 74: It is redundant to mention Nature in Norway again here, as it is in the first paragraph.

**Response: Yes. We will remove it.**

AR#1: It would be useful to know the percent rain vs. snow for each of these sites in the site descriptions.

**Response: We agree and will include an estimate of the percentage of rain vs. snow in the revised manuscript.**

AR#1: Figure 2 and 6: I appreciate the consistency of color throughout the figures for each site. However, for Figures 2 and 6, since the purpose is comparing years within each plot rather than sites, I would recommend each year in a different color (and line style) to help distinguish the years. This is not critical but would improve readability.

**Response: Thank you for pointing this out. We agree that figure readability is important and will update figure 2 and 6 to make it easier to distinguish between years. To keep the color consistency for each site, we suggest using different shades of each "site color" for different years, in addition to different lines styles.**

AR#1: Line 167: "according to its usual definition" – Please explain this or at least put a reference to how this is calculated

**Response: We agree and will include the equation used.**

AR#1: Line 199: Why was a KMO value of 0.5 used - Reference?

**Response: The KMO value of 0.5 was chosen to exclude data unacceptable for factor analysis. Most sites/seasons used in the factor analysis had KMO-values $> 0.7$, however a few sites had lower values in the snow-covered season. We will include the reference to Kaiser and Rice 1974, Educational and Psychological Measurement 34: 111-117. Further, we will include a table of the KMO-values for each site and season in the supplementary material.**

AR#1: Line 217: What are the various vegetation heights?

**Response: We will include information on vegetation height in the revised manuscript.**

AR#1: Figure C7: Hourly evaporation vs. cumulative growing degree day –
is this a needed figure? Not sure what value it adds and it is not adequately
discussed.

**Response: The reason for including figure C7 was to show that, although our evaporation model has constant vegetation parameters, and we know that the site vegetation changes through the growing season, we see no clear patterns in the deviation between modelled and observed evaporation with growing degree day. This indicates that the seasonal vegetation development had limited influence on the total evaporation. The link between the figure and our arguments about vegetation in the discussion will be made more clear in the revised manuscript.**

**Technical Corrections**

**We will update the manuscript to comply with the technical corrections suggested by AR#1**

---

## Author Comment (AC2)

**Response to comment by Anonymous Referee #2**

Thank you for the constructive review. Hereby, we would like to respond to your comments. In the following, the comments from the reviewer (AR#2) are in plain text, and our responses are in bold text. As the comments were embedded in an annotated version of the manuscript, we have extracted the comments and, where relevant, added line number and context in brackets.

**General comment**

AR#2: Please see my comments embedded in the attached annotated manuscript. This is a very nice paper. My suggestions are minor in nature and are meant to provide some small improvements to the paper to improve clarity for the reader and place the work in a broader context of the existing literature.

**Response: Thank you for the positive comment.**

**Abstract**

AR#2: [Line 6, re. controls on evaporation] Perhaps state temporal scale (i.e., seasonal).

**Response: We will update the sentence to: "We found that ecosystem evaporation was indeed mainly controlled by atmospheric evaporative demand, both in the snow-free and the snow-covered season, and that spring snow-cover duration impacted total annual evaporation."**

AR#2: [Line 7, re. soil moisture level] Be specific. "Never decreased to a level where it would restrict evaporation."

**Response: We will change the sentence to: "Soil moisture remained high during the measurement period, and our results suggest it never decreased to a level where it would restrict evaporation".**

**Introduction**

AR#2: This is a good well written paper. It could benefit from the results being put into the context of additional literature, notably from Canadian landscapes where several of these kinds of studies have been conducted. The authors already cite Nicholls, but could also do so from the perspective of environmental controls. Other authors/papers to consider would be Rouse, Leljedahl, Sonnentag, Lafleur and Spence. Here are a couple of examples to draw from: Rouse et al. 2000. Phys Geog. 21: 345-367, Liljedahl et al. 2011. Biogeosciences 8: 3375-3389, Warren et al. 2018. Ecohydrology 11: e1975, Lafleur and Rouse, 1988. Boundary Layer Meteorology 44: 327-347., Spence and Rouse, 2002. J. Hydromet. 3: 208-2018.

**Response: Thank you for the suggested literature. We found the suggested papers of Warren et al. (2018), Spence and Rouse (2002) particularly relevant and will discuss our results in the context of these studies. The paper of Liljedahl et al. (2011) was already included in the preprint (sections 1 and 4.1).**

AR#2: [Line 19] Not necessarily so in some continental locations (NW Canada).

**Response: We will rewrite the sentence to include exceptions, citing references such as Wang et al. 2013. Hydrol. Earth Syst. Sci., 17: 3561–3575. In addition, we will search for literature showing exceptions in other northern latitude areas.**

**Methods**

AR#2: [Line 96] was

**Response: Thank you. We will change the tense, from *is* to *was*.**

AR#2: [Line 119] Maybe introduce the term Eobs in this section somewhere so that the reader knows what is being compared to Epm when that is introduced below.

**Response: We agree, however, since the term $E_{obs}$ is used to describe observed evaporation only in the context of calculating the relative error, and *observed evaporation* is used elsewhere in the text and figures, we feel it could confuse more than it clarifies.**

AR#2: [Line 158, re. soil heat flux] This is an important term as it is used to derive available energy. But its description is a bit lacking. From Equation 2, it seems as if SHF = G. More detail is needed of the estimate of SHF.

**Response: Thank you for pointing this out. We suggest adding the following to section 2.2.2 Measured ancillary local data. "To estimate the soil heat flux (SHF) in the surface energy balance (equation 2), we used measurements from soil heat flux plates (Huskeflux), available at each site. Further details about measured ancillary local data can be found in Pirk et al. 2023 (Finse), Pirk et al. 2024 (Iskoras), Pirk et al. 2017 (Adventdalen) and Bekken et al. Carbon dynamics of a controlled peatland rewetting experiment in the Norwegian boreal zone *Scientific Reports* (in press) (Hisåsen).**

**Discussion**

AR#2: [Line 380, regarding level of soil moisture] See comment above.

**Response: We will change the sentence to: "Therefore, it is likely that the soil moisture content did not decrease to a level where it would restrict evaporation in the measurement periods."**

AR#2: [Line 398] Including a table of energy balance closure early on in the Results would be a good idea.

**Response: We agree and will include it.**

AR#2: [Line 425-428] While these are good points; I might argue that since available energy (driven much by incoming solar radiation) is a major control and the sites were not moisture limited, the seasonal ET rates are energy limited - which makes sense given that the authors found available energy to be very important. These sites evapotranspire at their maximum rates bounded by energy, and there was not much variation among the sites - or at least three of them. So precipitation controls the ET/P ratio. I would not expect the ET/P ratio to be the same among the sites. Maybe it would be best to say this more explicitly.

**Response: We agree and suggest adding the following sentence in line 425: "The evaporation ratio is thus mainly controlled by precipitation."**